# Variation Patterns of Forest Structure Diversity after Set-Aside in Rarău-Giumalău Mountains, Romania

Cătălina-Oana Barbu [1,2], Gabriel Duduman [1,2,3,*] and Cezar Valentin Tomescu [1,3]

1    Forestry Faculty, "Stefan cel Mare" University of Suceava, Universitatii Street 13, 720229 Suceava, Romania
2    Applied Ecology Laboratory, "Stefan cel Mare" University of Suceava, Universitatii Street 13, 720229 Suceava, Romania
3    Forest Biometry Laboratory, "Stefan cel Mare" University of Suceava, Universitatii Street 13, 720229 Suceava, Romania
*    Correspondence: gduduman@usv.ro; Tel.: +40-230-522-978 (ext. 564)

**Abstract:** The present study aims to analyze the set-aside effect on the current structure diversity of mountain temperate forests from the Natura 2000 site Rarău-Giumalău. In the past 80 years, the area of entirely protected forests successively increased to up to 77% of the site. The description of past structure diversity was based on the analysis of management plans drawn up for these ecosystems after 1940, while their current structure diversity was based on a tree census carried out in 2015. The forests' structure diversity was described in relation to: tree dimensional heterogeneity; wood volumes homogeneity of the living trees throughout the site; variability of the standing and lying dead wood volume; number and basal area of large trees; natural regeneration. The results show that forest stands where no harvest has ever been registered record the highest level of tree size heterogeneity, while in previously managed forests, the current structure diversity was influenced by the harvesting intensity. The dimensional diversity of trees also depends on the structure, density and age of forest stands at the moment when they are set aside. We observed that the volume of dead wood on the ground greatly increases after abandonment of timber production and that there is a progressive decrease in the number and percentage of large trees in the first 40 years after the last timber harvest, accompanied by a significant decrease in living trees volume. Nevertheless, the number of large trees in stands where the last timber harvesting occurred more than six decades ago is 1.8 times higher than that of the corresponding number in stands where no harvesting was ever performed. The time elapsed since the last harvest generated important changes in the regeneration process, which seems to stabilize after three decades. The forest stands' reaction after set-aside very much depends on their characteristics at the time of exclusion from timber production, especially their age and structure. After 80 years since set-aside, the ecosystem processes and descriptors begin to look very much like those in the forests unaffected by human actions, but the old-growth characteristics have not entirely recovered.

**Keywords:** Natura 2000; forest management history; set-aside forests; forest structure diversity; old-growth forest; Slătioara UNESCO site

## 1. Introduction

In recent decades, biodiversity has become one of the main topics discussed when it comes to forest management and forest conservation [1–3]. While globally forests cover an area of approximately 4 billion hectares and hold 80% of the biodiversity, losing biodiversity has substantial consequences for the proper functioning of forest ecosystems [4]. It is well known that forests offer a wide range of ecosystem services [5], and increasing the anthropogenic pressure directly impacts the biodiversity and, thus, the provision of these services [6]. Maintaining and preserving forests' biodiversity is commonly based on two major strategies: (i) segregation strategies, where protected and production areas are spatially segregated and (ii) nature-based silviculture strategy, which promotes the forests'

multi-functionality. Such forest management decisions and strategies are usually derived at a national level [7,8], considering the crucial role played by the forest management in driving the forest structure and composition, and consequently the profound effects on biodiversity and functioning of forest ecosystems [9].

In Europe, during the past two centuries, the biodiversity of native forests has been altered by management of differing intensities [10]. According to Forest Europe 2020, 75% of the forests are even-aged and only 25% uneven-aged [11]. Today, primary forests represent less than 1% of European forests [12]. In Eastern Europe, where substantial old, near-natural forests exist [12], a wide variety of approaches have been implemented, aiming to prevent further loss of structural and functional diversity. One of these approaches consists in the designation of strict forest reserves or set-aside forests [8]. Set-aside forests are defined as lands covered by forests primarily managed for the purpose of nature conservation [13]. These forests, left to free development, have greatly contributed to the maintenance and recovery of old growth characteristics [14–16], but in stands with an intensive management type, the recovery of the old growth characteristics after abandonment is a slow process [12].

Some authors [17,18] consider that native biodiversity is adapted to unmanaged forests, as it has grown and developed under a regime of natural disturbances. Therefore, the biodiversity of forests set aside from forestry is often considered best preserved by non-intervention [19]. Other authors [20,21] consider that in European intact forests the current biodiversity is a consequence of past human disturbances. Thus, in some of the existing reserves, past silvicultural interventions have led to changes in forest habitat with consequences for biodiversity [3]. When we refer to managed forests, we see that the biodiversity is certainly influenced by silvicultural interventions. The effect of the silvicultural treatments on tree species diversity and tree size heterogeneity is determined by the intensity of the treatment [9,22]. Management also changes the availability of deadwood with direct effects on biodiversity [23], because the deadwood volume is used in Europe as an important indicator of forest biodiversity [24] and, in the context of biodiversity evaluation [25–27], it is an important measure of old-growth characteristics and naturalness [28].

Hence, studying the past forest management is a key element in understanding biodiversity-related changes and therefore in adapting management strategies to current challenges associated with forests. Forest management histories differ between European countries. For example, there are inconsistencies when we compare managed and unmanaged stands across different regions [29]. In the last decades, there has been an increased interest in the interplay between the past management, forests' diversity [1–3] and the time that has elapsed since forests were set aside [30], especially because the set-aside forests boost above-ground carbon stocks and plant diversity [31]. These are the important reasons considered by the Aichi Biodiversity Target 11 (under the auspices of the Convention on Biological Diversity) to promote setting aside vast (currently managed) areas for conservation purposes [32]. However, setting aside new areas for conservation seems to be bureaucratically obstructed in some European countries and, therefore, it will be difficult to achieve the international commitments [13]. Thus, the areas already set aside are becoming more valuable to our understanding of how the ecosystems react when anthropogenic pressure disappears and how to quantify the rhythm and intensity of natural changes.

Most of the set-aside forests in Europe are included in the EU Natura 2000 network of protected areas. However, some authors proposed that, in many of these forests, the habitat conditions are not better than in unprotected areas [33]. So far, several studies have highlighted the important role of management in European forests in mountainous areas, by comparing diversity in managed and unmanaged forests [14,34]. For example, it is known that, in the case of pure Norway spruce stands, forest management has a greater impact on tree diversity when compared to mixed mountain forests, because the intensively managed forests are characterized by a low diversity and low resilience to disturbing factors such as wind, snow, fire or insects [35–39]. Thus, historical data collected from intact forests and set-aside forests are becoming extremely valuable for establishing appropriate

management measures for managed forests with low diversity, in order to respond better to the current global challenges. Some researchers [40] investigated the impact of past forests management on plant diversity using as a starting point the historical evidence in European beech forests. Other studies investigated the biodiversity response to forest management at the stand level [41] and landscape level [42]. Even though the forest diversity is driven by the time elapsed since the abandonment of management, the results are context-sensitive above all to management history [43,44].

Due to the fact that the impacts of forest management on the biodiversity are not fully understood, a more complete synthesis is needed to provide strong evidence, on a case-by-case basis, in order to figure out how various management types applied in the past may have affected the structure diversity in forests set-aside for conservation. Our preliminary hypotheses were that: (i) the recovering process of old-growth characteristics depends on the management type and the harvesting intensity applied before set aside; (ii) the main structural characteristics of forest stands recorded at the moment of set-aside define the rhythm of the recovery process; (iii) seven decades of an entire protection regime are not sufficient to entirely recover the old-growth characteristics.

Thus, our study aims to analyze the impact of previous forest management on current forest structure diversity in the Natura 2000 site ROSCI0212 Rarău-Giumalău, Romania, based on the fact that the forest stands within this protected area were set aside at different moments. The specific objectives of this study are: (i) to establish if the type of past forest management is responsible for the current structure diversity in set-aside forest stands from the Natura 2000 site Rarău-Giumalău; (ii) to evaluate how the main characteristics of the forest stands at the moment of set-aside contributed to the current tree dimensional diversity; (iii) to identify the variation patterns of tree diversity-related forest characteristics after the last harvest; (iv) to determine if the forest stands entirely recovered the old-growth characteristics after set-aside.

## 2. Materials and Methods

### 2.1. Study Location

This research was carried out in the Natura 2000 site ROSCI0212 Rarău-Giumalău (N2000-RG), an area established in 2009 (proposed in 2007) and designated for the protection of species and habitats of community interest. The site is located in north-eastern Romania, in the north of the Oriental Carpathians (the Rarău and Giumalău Mountains), at altitudes between 800 and 1720 m above sea level. It has a total area of 2546.9 ha and is divided into two continuous zones (Figure 1): the Giumalău area to the west (341.05 ha, of which 99.3% are forest habitats) and the Slătioara-Rarău area to the east (2205.85 ha, of which 78% is represented by forests).

The boundaries of N2000-RG have been set to include the five existing nature and scientific reserves in the region, with a total area of 2424.31 ha (Table 1). The 6.0 ha area corresponding to the scientific reserve "Peștera Liliecilor" is below ground and is located within the "Rarău—Pietrele Doamnei" Nature reserve. In 2017, the reserves "Codrul secular Slătioara" and "Fânațele montane Todirescu" were included on the UNESCO World Heritage List, with a core area of 609.12 ha and a buffer area of 429.43 ha [45]. Forests represent the dominant land use type (81.2%), followed by pasture lands and forested pastures (17.0%), rock formations, partially covered with tree species (1.6%), while the rest of the area is represented by powerlines and construction (0.2%).

The dominant tree species are silver fir (*Abies alba* Mill.), Norway spruce (*Picea abies* (L.) H. Karst.) and European beech (*Fagus sylvatica* L.) in the Slătioara-Rarău area, while in the Giumalău area the main tree species is the Norway spruce. In the Giumalău reserve, the mean annual temperature is 1.6 °C, and the mean precipitation is about 973 mm, while in the Slătioara-Rarău area the mean annual temperatures vary between 3.8 °C at high elevations and 5.9 °C at low elevations, and the mean annual precipitation is between 700 and 810 mm [45].

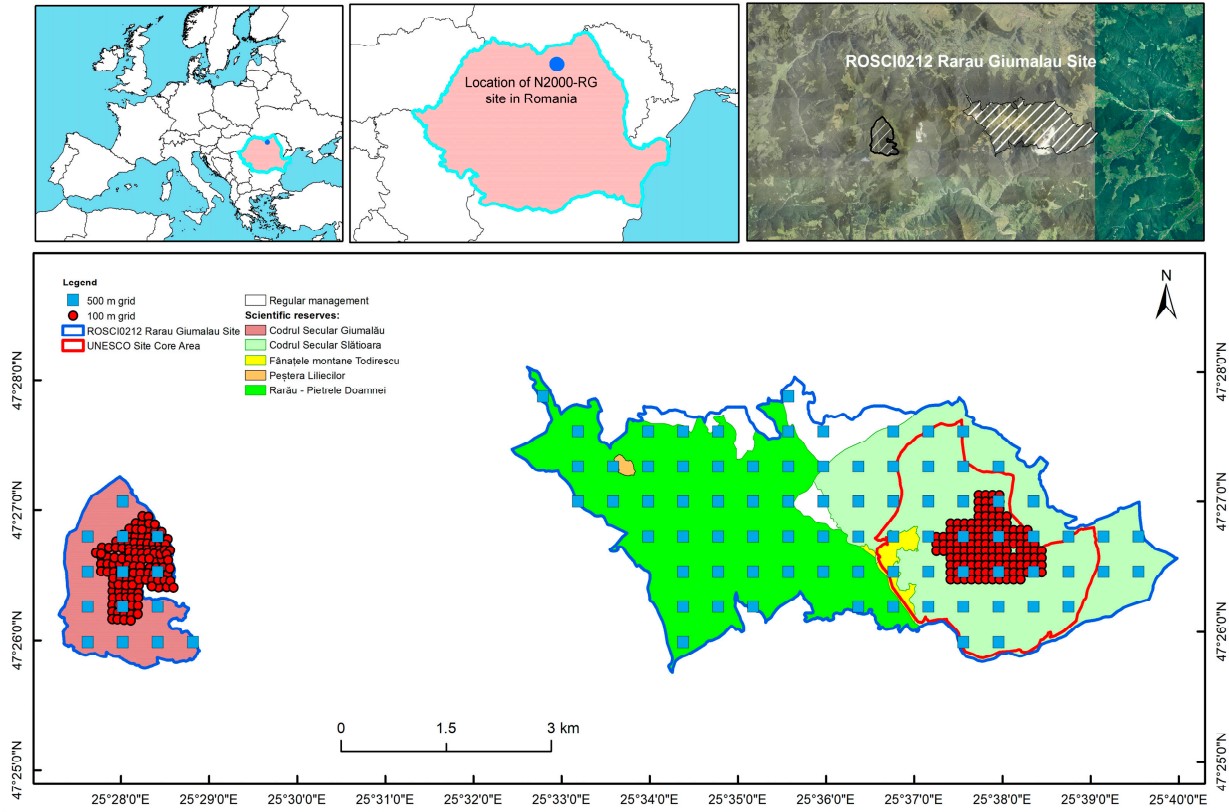

**Figure 1.** Location of N2000-RG site, reserves inside the study area and design of the research network (reserve boundaries from http://mmediu.ro/articol/date-gis/434, accessed on 6 July 2022). Maps were created using ArcGIS® software by Esri. ArcGIS® and ArcMap™ are the intellectual property of Esri and are used herein under license. Copyright © Esri. All rights reserved. For more information about Esri® software, please visit www.esri.com (accessed on 6 July 2022).

**Table 1.** Summary of reserves inside the study area.

| Name of Reserve | Type | Main Conservation Goal | Total Area (ha) |
|---|---|---|---|
| Codrul secular Giumalău | Nature reserve | Forest species and habitats | 338.81 |
| Codrul secular Slătioara | Nature reserve | Forest species and habitats | 1064.2 |
| Rarău—Pietrele Doamnei | Nature reserve | Geology, flora species, and forest habitats | 971.0 |
| Fânațele montane Todirescu | Nature reserve | Flora species | 44.3 |
| Peștera Liliecilor | Scientific reserve | Speleological habitats | 6.0 |

In 2015, a permanent research platform (PRP) was established, overlapping the entire area of the site, aiming for the monitoring of species and habitats. At the level of the Slătioara-Rarău area, this network has been described in the literature [45]. The PRP is structured on two levels: The first level of the PRP (labelled 1-PRP) corresponds to a 500 × 500 m square grid, overlapping the entire N2000-RG; The second level of the PRP (labelled 2-PRP) corresponds to a 100 × 100 m square grid, and resulted from the increased density of the 1-PRP [45] in the areas previously described in the literature as having higher ecosystem complexity [46–48].

The PRP consists of 347 circular permanent sample plots (SPs), of 500 square meters each, out of which 95 belong to 1-PRP and 252 to 2-PRP (Figure 1). Due to the fact that the 2-PRP overlaps entirely on forest stands never subjected to wood harvesting, in order to avoid influencing the results by spatial autocorrelation of the very large number of SPs installed in the core areas of the Slătioara and Giumalău reserves, we based our analysis only on the SPs from 1-PRP. Two SPs from 1-PRP are currently outside the study area, due to an adjustment of the borders of N2000-RG which occurred in 2017, compared to the

previous border. These plots were inventoried and were not excluded from the initial list of SPs, considered as control plots of the nearby Natura 2000 site [45]. Due to administrative constraints, for six plots in the Giumalău area the measurements were not performed in 1-PRP, and were therefore not included in this analysis. In addition, the plots installed in habitat types other than forests were excluded from the present analysis. Thus, 70 SPs were considered in this study.

### 2.2. Overall Description of Past Forest Management in N2000-RG

The largest part of N2000-RG overlaps nature and scientific reserves. As the "Peștera Liliecilor" Scientific Reserve and "Fânațele montane Todirescu" Nature Reserve are not subject to forest management, the study of the history of forest management will refer only to the forests within the site, in particular to forest reserves. In order to meet the research objectives, through the analysis of the forest management plans (FMPs) (Table S3) we obtained information on the time the forest compartments were set aside and on the dynamics of the areas managed as entirely protected after 1940 (Figure 2).

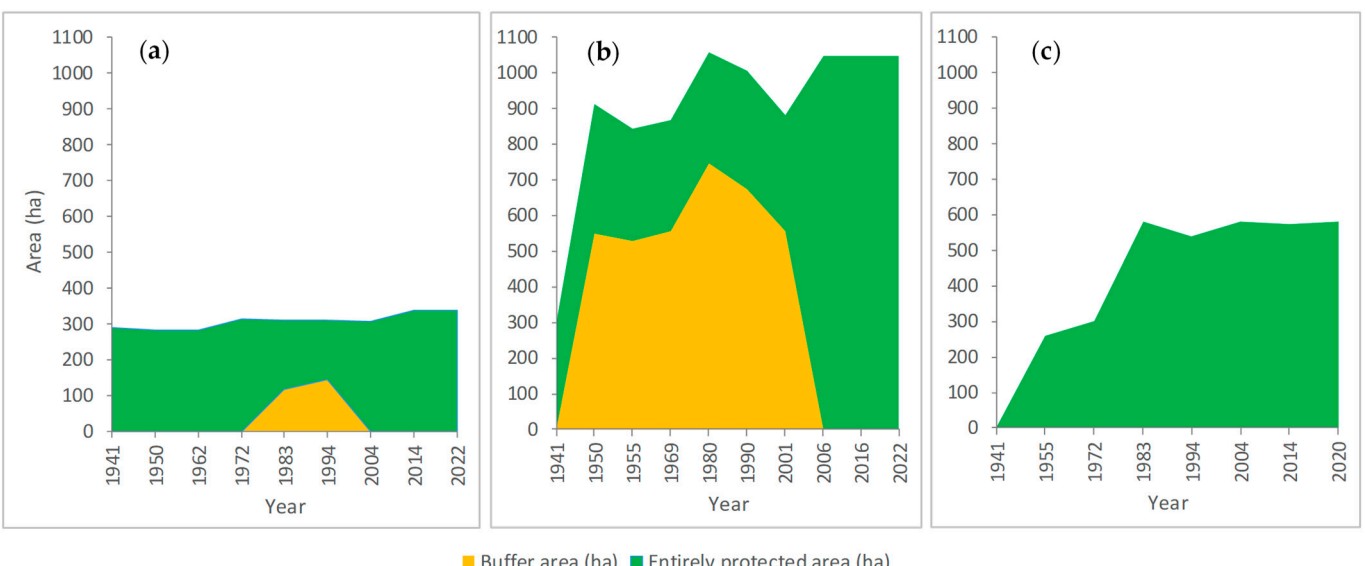

**Figure 2.** Dynamic of forest areas set aside since 1940 in N2000-RG: (**a**) Codrul secular Giumalău; (**b**) Codrul secular Slătioara; (**c**) Rarău—Pietrele Doamnei Nature reserve.

The first attempts to designate Slătioara Forest as a reserve date back to 1904–1906, due to the scientific importance of the area in terms of flora and forestry. In 1921, the proposal by Professor Mihail Gușuleac to establish Slătioara forest as a reserve was taken into account, and the FMP of 1925 describes Slătioara nature reserve with an area of 671.11 ha, as stipulated in decision 46662/1925 [49]. Pressure was put on the harvesting of the forest and, in 1930, in the new FMP, this area was no longer classified as a reserve while in 1933, at its eastern boundary, the first felling took place. In 1934, Gușuleac established a nature protection commission and halted the exploitation. He proposed the forest as a reserve for an area of 295.28 ha, but only in 1941, by the Decision of the Council of Ministers no. 1294, was the area officially established as the reserve of the Romanian Academy [49]. In 1950, a buffer area of 360 ha was established, which was maintained until 2006, when it was included in the strict protection regime.

The area of 42.5 ha belonging to "Fânațele montane Todirescu" was established in 1941 as a natural reserve of floristic interest, as part of the Slătioara reserve, included at that time in the national forest fund. In 1954, it was removed from the forest fund and transferred to the zoo-technical service [50] and, in 1980, its re-measured area was 44.4 ha [51], very close to the current one of 44.3 ha [52].

In "Codrul secular Giumalău", the strict protection regime was adopted in 1937 through the forest management plan [53], but the reserve was officially established by the Decision of the Council of Ministers no. 9942 of 1941 [54], with an area of 290.62 ha [55,56]. In this case, part of the reserve was transformed in the buffer area from 1983 to 1994, due to the damage produced by wind in some forest compartments and to formal decisions to extract the damaged wood. Similar to the case of Slătioara reserve, low-intensity cuts were allowed in the buffer areas, aiming to isolate the core area from external influences. The "Rarău—Pietrele Doamnei" was established in 1955 (Decision no. 1625 of the Council of Ministers) as a Reserve of the Romanian Academy, with an area of 258.5 ha [57]. Currently, the entire area of strictly protected forests in N2000-RG is 1961.82 ha.

Within the analyzed period, according to the FMPs provisions and monitoring records of the forest districts, forest management (in areas outside the entirely protected zones) was characterized by low- or very low-intensity cuts. The close-to-nature forestry (usually a single-tree selection system) was applied in the forest stands aged within more than 100–120 years from the buffer areas of the entire N2000-RG. The harvesting intensities per decade were less than 10% of the standing volume. In young even-aged and two-aged stands, commercial and pre-commercial thinning was applied, in average and moderate intensities, depending on the stand age and forest type. In production forests, clear cuts were performed on very small areas, but not in the past 50 years. Accidental and sanitary cuts are more frequently recorded in monitoring files. Accidental cuts aimed at extracting the fallen trees (damaged by wind or snow) from the buffer and production areas, and usually the harvested wood volumes varied between 1 and 3 $m^3 \cdot yr^{-1} \cdot ha^{-1}$, exceptionally reaching values of 5 to 7 $m^3 \cdot yr^{-1} \cdot ha^{-1}$. Sanitary cuts aimed at preserving a good healthy state of the forest stands in the buffer and production areas were also performed, with harvested volumes usually lower than 1 $m^3 \cdot yr^{-1} \cdot ha^{-1}$.

### 2.3. Methods

#### 2.3.1. Data Collection

Two types of data were collected: field data to evaluate the current state of the forest ecosystems, and historical data to describe past forest management. The method used for data collection in the field is described in detail by Duduman et al. [45] for living trees, standing dead trees, dead wood on the ground and natural regeneration. The same protocol was used for all 70 plots considered in this study, within the inventoried area (3.9 ha). The natural regeneration was evaluated in four subplots, each of 3.14 square meters, installed in every plot.

In the case of standing trees (living and dead), the following characteristics were used in the analysis: species, dbh (cm) and total height (m). For dead wood on the ground we considered the species, length of the log within the SP (m), diameters at the ends (cm) within the SP and the decay class according to Maser et al. [58]. In the case of natural regeneration, the number of saplings at the SP level was taken into account.

Past forest management in the studied area consisted of bibliographical research, mainly based on the analysis of the FMPs. All available FMPs corresponding to N2000-RG were explored (27 analyzed FMPs), considering the moment since the first areas in the current site were set aside. During the past 90 years there were many changes in terms of forest administration and, for this reason, the first step of our analysis was to identify the forest district (FD) and forest management unit (FMU) each SP belonged to at a specific moment, and to extract data from the proper FMP (Table S1 from Supplementary Materials). Our bibliographical analysis was conducted until 2015, the year in which data were collected in the field from the PRP.

The second step was to overlap the SPs network on the forest maps, to identify the forest compartment corresponding to each SP and then to extract data from FMPs at the forest compartment level. The following data were taken from the FMPs: compartment number, area (ha), management type applied before last harvesting operation (no human intervention, uneven-aged management, even-aged management), year when included in

strict protection regime, year and type of last harvesting operation, past forest management intensity, and stand characteristics at the moment of set-aside: stand structure (even-sized; two-sized; uneven-sized irregular; and uneven-sized balanced), mean age and density. For each forest compartment over which an SP overlaps, we extracted from forest management plans and summed up the wood volumes harvested between 1941 and the year of the last harvest. We then computed the intensity of past forest management by dividing this volume by the mentioned time frame and by the area of the forest compartment in the year of the last harvest. Stand density is quantified in FMPs based on a scale between 0 (trees are lacking) and 1 (the projection of tree crowns to the ground covers entirely the stand area), with a 0.1 step. Our analysis was based on data collected at the forest compartment level for the entire studied period, as the FMPs are the only sources for historical data related to forest management in the studied area. Information on the precise location of the cuttings inside the forest compartments does not exist in the forest districts archives, so these historical data were indirectly connected to our plots, based on spatial locations of SPs and forest compartments.

### 2.3.2. Data Processing

The database obtained from the data collection phase was further improved with processed data, based on the inventory carried out in 2015 in the plots from the 1-PRP. In order to assess the variation patterns of current forest attributes with respect to time since the last harvest occurred, we analyzed every SP closeness to old-growth forests [59], considering the following characteristics: variation in tree sizes, above-ground wood volume of living and dead trees, volume of dead wood on the ground and its distribution on decay classes, number and basal area of large trees, and presence of natural regeneration.

Thus, the tree size heterogeneity was assessed at the plot level using the Gini index, which ranges between 0 and 1 [38,60]. This index is recommended in the literature for the assessment of forests' structural diversity [61–63] and is computed with the following formula [63,64]:

$$G = 1 - \sum_{i=1}^{k} [(ba_{i-1} + ba_i) \cdot (n_i - n_{i-1})] \tag{1}$$

where $ba_i$ ($ba_{i-1}$) is the cumulative fraction of the basal area (%) of the trees from all diameter classes thinner than or equal to the *i*th ($i - 1$) diameter class (for $i = 1$, $ba_{i-1} = 0$); $n_i$ ($n_{i-1}$) is the cumulative fraction of the number of trees (%) from all diameter classes thinner than or equal to the *i*th ($i - 1$) diameter class (for $i = 1$, $n_{i-1} = 0$); and $k$ represents the number of 2 cm-diameter classes.

The wood volume of each standing tree (living or dead) was quantified [64] with the logarithmic equation established in the literature [65], and its specific coefficients $a_{0i}$, $a_{1i}$, $a_{2i}$, $a_{3i}$, $a_{4i}$ [64] (Table S2) for the *i*th species:

$$\log_{10} vu_{i;j} = a_{0i} + a_{1i} \cdot \log_{10} (dbh_j) + a_{2i} \cdot \log_{10} (dbh_j)^2 + a_{3i} \cdot \log_{10} (h_j) + a_{4i} \cdot \log_{10} (h_j)^2 \tag{2}$$

where $vu_{i;j}$ is the single tree above-ground wood volume; $dbh_j$ represents the diameter at the breast height of the *j*th tree (with a *dbh* threshold of 5 cm); $h_j$ is the total height of the *j*th tree.

The lying dead wood volume was computed for every log using the cylinder formula for volume, considering the length of the log within the SP limits (cm) and the diameters at the two ends of the log inside the SP (with a threshold of 5 cm at the thick end). The identification of large trees was carried out considering a dbh threshold of 60 cm. For further data processing, discrete classes were distinguished with respect to time elapsed since the set-aside and time elapsed since the last timber harvesting occurred. Class size was established considering the dynamic of set-aside forests (Figure 2), which strictly depended on the moment the new FMPs were revised (every ten years). Thus, the size of the class was established to 10 years. The analysis of the differences between means computed for response variables (Gini index, volume of standing and lying dead wood,

volume of living trees, number and basal area of large trees, number of seedlings) on classes of explanatory variables (type of past forest management, intensity of past forest management, time since last timber harvesting, stand structure and density when set aside), was carried out using a PERMANOVA analysis [66] because all experimental distributions differ from the normal distribution (Shapiro–Wilk test), even after the data transformation $(x' = \log(x + 1))$. When significant differences were found, the Bray–Curtis test for multiple pairwise comparison was used to compare means [67]. In the case of continuous but not normally distributed data series, the correlation analysis was performed by means of the Spearman test. The analyses were performed with XLSTAT-PRO 2012 (Addinsoft, New York, NY, USA), plugged into EXCEL 2016 (Microsoft Corp., Redmond, Washington, DC, USA) and with PAST 4.11 (Natural History Museum, University of Oslo, Oslo, Norway).

## 3. Results

### 3.1. The Effect of Past Forest Management on the Current Forest Structure Diversity

Current forest structure diversity significantly depends ($F = 10.9$; $p = 0.0001$) on the forest management type applied at the compartment level before the last timber harvesting (Figure 3). The mean value of the Gini index in the case of forests where uneven-aged management was applied in the past (0.534) is much closer to that of forest stands with no human intervention (0.641), compared to the mean value of the index recorded in the case of even-aged management (0.387).

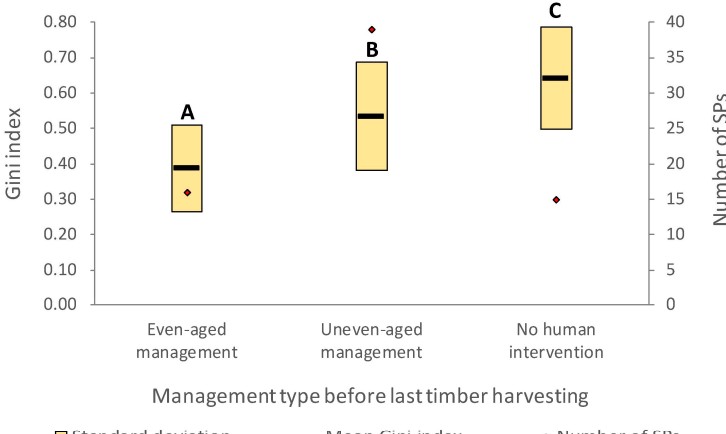

| Pairwise p-values | No human intervention | Uneven-aged management | Even-aged management |
|---|---|---|---|
| No human intervention | | 0.0356 | 0.0001 |
| Uneven-aged management | 0.0356 | | 0.0015 |
| Even-aged management | 0.0001 | 0.0015 | |

**Figure 3.** Forest structure diversity on forest management types (green color highlights the significant differences: $p < 0.05$). A,B,C—different letters indicate significant differences.

While the intensity of past forest management shaped the current structural diversity of the forests, our findings show that the highest structural diversity is recorded by stands from which wood has never been harvested and that the value of the Gini index decreases in the stands where harvesting intensity was higher in the past (Figure 4). However, the differences recorded between the classes of harvesting intensity are not significant ($F = 0.7281$; $p = 0.6739$), mainly due to the low number of plots in high-intensity classes, but also due to the high standard deviation of the Gini index registered in low-intensity classes.

If structural differentiation thresholds are considered [63], in N2000-RG the uneven-sized balanced structures are found mainly in the plots with no harvest in the past (mean G = 0.604) or where harvesting intensity was lower than $1.0 \, \text{m}^3 \cdot \text{yr}^{-1} \cdot \text{ha}^{-1}$ (mean G = 0.518), while in all the other SPs the forest stands frequently present uneven-sized irregular or two-sized structures.

The dimensional heterogeneity of the trees remains relatively constant in the first 40 years after the latest harvest, decreasing 41 to 60 years later. The highest level of tree dimensional heterogeneity was found in stands where at least 60 years had elapsed since the last intervention (mean Gini = 0.612), very similar to the values determined in stands

from which timber has never been harvested (mean G = 0.604). However, the differences between means are not statistically significant ($F = 1.091$; $p = 0.3778$), except those between time classes of 21–30 and 41–50 years (Figure 5).

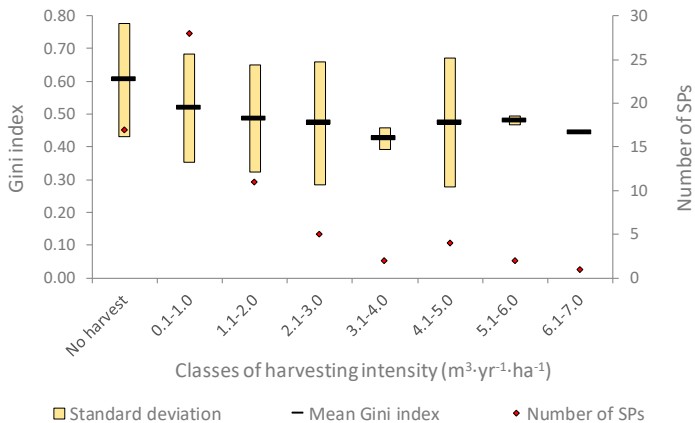

**Figure 4.** Effect of past harvesting intensity on current forest structure diversity.

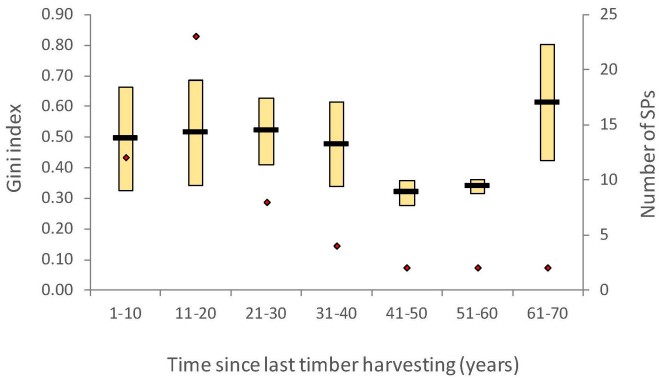

| Pairwise p-values | 1-10 | 11-20 | 21-30 | 31-40 | 41-50 | 51-60 | 61-70 |
|---|---|---|---|---|---|---|---|
| 1-10 | | 0.8246 | 0.5274 | 0.9256 | 0.1913 | 0.2218 | 0.3767 |
| 11-20 | 0.8246 | | 0.6355 | 0.7643 | 0.1132 | 0.2054 | 0.5351 |
| 21-30 | 0.5274 | 0.6355 | | 0.518 | 0.0426 | 0.0664 | 0.3686 |
| 31-40 | 0.9256 | 0.7643 | 0.518 | | 0.257 | 0.274 | 0.3366 |
| 41-50 | 0.1913 | 0.1132 | 0.0426 | 0.257 | | 0.6682 | 0.3331 |
| 51-60 | 0.2218 | 0.2054 | 0.0664 | 0.274 | 0.6682 | | 0.3318 |
| 61-70 | 0.3767 | 0.5351 | 0.3686 | 0.3366 | 0.3331 | 0.3318 | |

**Figure 5.** Trees' dimensional heterogeneity on time classes since last timber harvesting (green color highlights the significant differences: $p < 0.05$).

The 1-PRP also includes 20 plots installed in stands that were never subjected to the entire protection regime, and here the Gini heterogeneity index varies from 0.23 to 0.74, with an average of 0.54. Sixty percent of these plots, with recorded Gini values higher than 0.5, correspond to the uneven-sized balanced structures [48,63]. A deeper analysis at the SP level revealed that all these SPs with Gini index values higher than 0.5 are found in forest stands included since 1941 in the buffer zones of the protected areas, stands in which interventions were minimal and oriented only to maintaining the proper health of the stands. For this reason, we found a high heterogeneity of tree sizes in the class of SPs never subjected to a strict protection regime.

It was also found that there are stands where the last forest operation has been recorded in the past 20 years, but their dimensional heterogeneity is high, as well as stands in which the interventions ceased 41–60 years ago, but whose dimensional heterogeneity is reduced.

### 3.2. Stand Structural Characteristics When Set-Aside Determines the Current Tree Size Heterogeneity

To better explain these results, we extended our analysis by taking into account the main characteristics of the forest stands at the moment they were set aside (Figure 6). We noticed the following:

1. Current dimensional heterogeneity of trees significantly depends on the structure of the stands at the moment they were set aside (Figure 6a). The Gini index reaches the highest values in the case of stands that were already uneven-aged balanced (mean G = 0.6770) or irregular (mean G = 0.5423).

2. The highest structure diversity is registered in forest stands with densities between 0.5 and 0.7 at the moment they were set aside. These stands significantly differ if compared with stands that had registered densities of 0.8 or 0.9. The variation pattern of tree dimensional heterogeneity indicates that the highest current structure diversity occurs in the case of stands with low densities at the moment they were set aside (Figure 6b). Stand densities between 0.5 and 0.7 favored natural regeneration, which later contributed to the trees' dimensional differentiation.

3. Stands' ages at the moment they were set aside significantly influenced the current dimensional heterogeneity of trees (r = 0.5771, *p* < 0.0001). The trend is positive (Figure 6c), being related to both the structure and density of the stands in the past.

A deeper analysis indicates that the forest stands from which trees were harvested in the past 20 years registered at the time they were set aside a high structural diversity, high densities and old ages, and that the forest operations performed had low intensities, specific to the buffer zone of which they were part. Part of the stands from where the last harvest was registered 41 to 60 years ago had at the time of setting aside younger ages and simplified structures (two-sized stands), which explains their current low dimensional heterogeneity of trees (Figure 5).

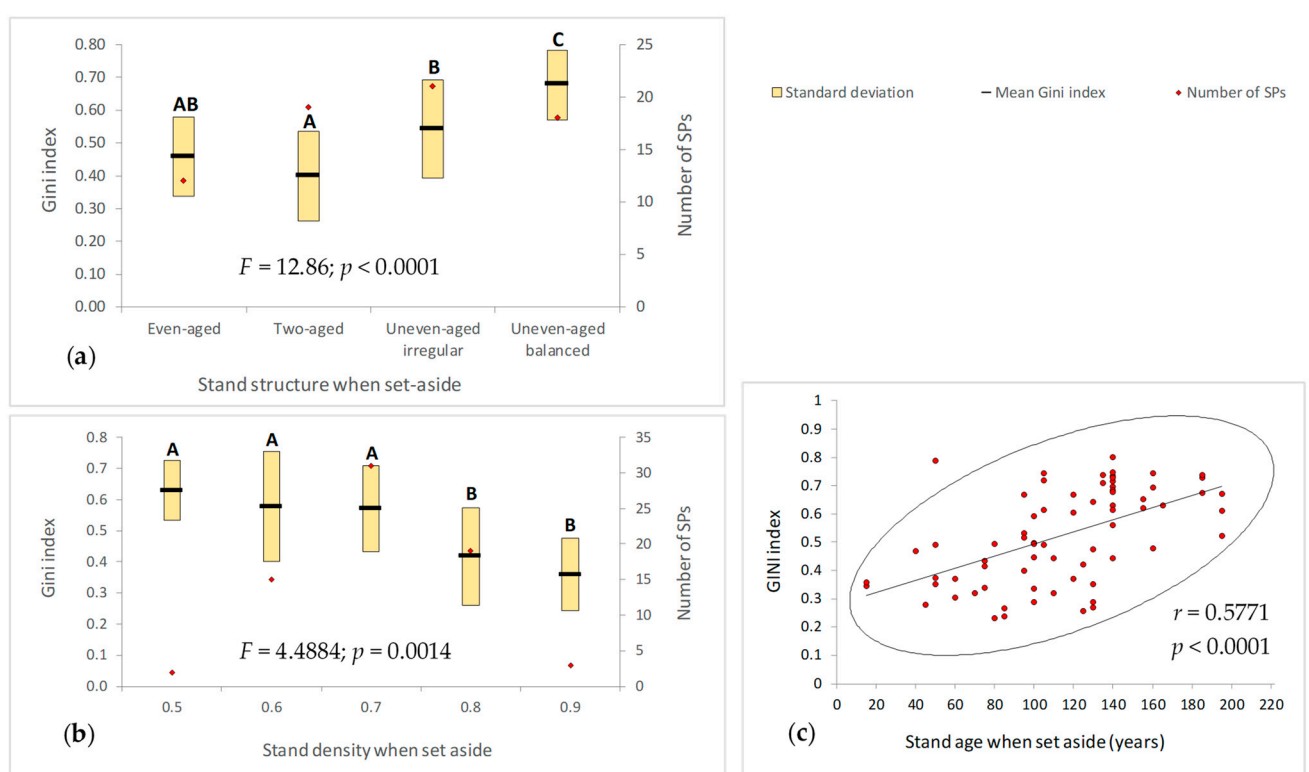

**Figure 6.** Trees' dimensional heterogeneity against the characteristics of the forest stands at the moment they were set aside: (**a**) structure; (**b**) density; (**c**) age. A,B,C—different letters indicate significant differences.

### 3.3. Variation Patterns of Forest Stands Attributes after Last Timber Harvesting

3.3.1. Dead Wood Volume after Forests' Set-Aside

As a result of halting the timber harvests, the mean volume of standing dead trees began to increase slightly. In the first decades after the last harvest, mainly small trees

were naturally eliminated and, therefore, the mean volume of standing dead trees was smaller. Subsequently, trees from the upper stand layer also died, with the mean volume of dead trees reaching a maximum (60.1 $m^3 \cdot ha^{-1}$) 31 to 40 years after the last wood harvest (Figure 7a), which is much higher than the value recorded in stands where wood had never been harvested (41.0 $m^3 \cdot ha^{-1}$). Starting with the fifth decade, this volume began to decrease, most likely due to the stands' structural improvements and the balancing of the ratio between the volume of living trees and that of dead wood (both standing and on the ground). The differences between means are not significant ($F$ = 1.091; $p$ = 0.3805), excepting those between time classes 21–30 and 41–50 years ($p$ = 0.0445).

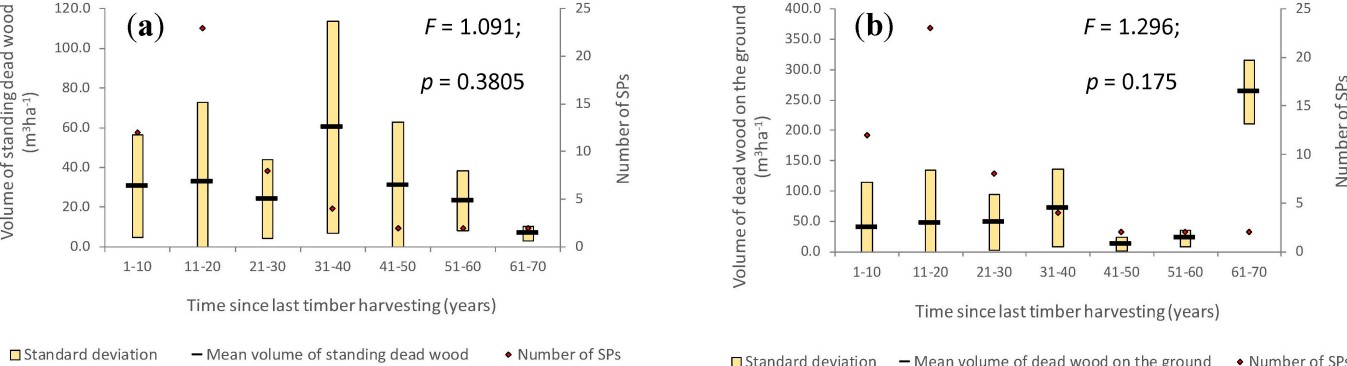

**Figure 7.** Volume of standing dead wood (**a**) and volume of dead wood on the ground (**b**) against time classes since last timber harvesting.

Dead wood volume on the ground registers significantly different values with respect to time elapsed since the last timber harvesting (Figure 7b). Excepting the SPs corresponding to forest compartments where the last timber harvesting was registered 41 to 60 years ago, which were very different from the other stands in terms of age and structure, the volume of dead wood on the ground greatly increases after the abandonment of timber production. In the case of forest stands where the last harvesting operation was registered seven decades ago, the mean volume of dead wood on the ground reached 263 $m^3 \cdot ha^{-1}$ ($SD = \pm 52\ m^3 \cdot ha^{-1}$), much higher than in the case of the stands where no human intervention was ever registered (110 $m^3 \cdot ha^{-1}$). Significant differences were found only between the seventh and the first decade ($p$ = 0.0227) and, respectively, the seventh and the second decade ($p$ = 0.0286).

In the case of stands from which no tree has been harvested in the past 61 to 70 years, the volume of dead wood on the ground in the fifth decay class (89 $m^3 \cdot ha^{-1}$) represents a third of the total volume of dead wood on the ground. Moreover, it is twice as high as the volume of dead wood on the ground recorded in the fifth decay class in the forest stands where no human intervention has ever been registered, which shows that the effect of abandoning the production regime was felt on the structure of these stands a few decades ago, shortly after they were set aside, which is also a result of the old age of these stands at that time (142 years on average). The standard deviation in the dead wood volume on the ground in the fifth decay class is 2.3 times higher in the stands set aside 61–70 years ago when compared to the stands with no timber harvest registered.

### 3.3.2. Above-Ground Wood Volume after Forests' Set-Aside

In the first 40 years after the cessation of timber harvesting, the volume of living trees per hectare decreased by 51% (Table 2), tree mortality increased and the total amount of dead wood became almost seven times higher after more than 60 years since the last wood harvesting. In the case of these stands, the mean dead wood volume per hectare (standing and lying) reached 270 cubic meters, 1.8 times higher than the mean dead wood volume founded in stands that were never affected by human intervention. This shows that the forest stands set aside go through a complex process of structural optimization, tending to

the balance between the wood volume of living trees and that of dead wood quantified, in the case of analyzed stands never affected by timber harvesting, by a ratio of about 4 to 1. If we distinguish between standing and lying dead wood, the ratio between the volumes of living trees, lying dead logs and standing dead trees becomes 14:3:1.

**Table 2.** Dynamic of wood volume with respect to time since last harvest.

| Time Since Last Harvest (Years) | No of SPs | Living Trees | | Above-Ground Wood Volume (Living Trees and Dead Wood) | |
|---|---|---|---|---|---|
| | | Mean Volume ($m^3 \cdot ha^{-1}$) | SD of Volume ($m^3 \cdot ha^{-1}$) | Mean Volume ($m^3 \cdot ha^{-1}$) | SD of Volume ($m^3 \cdot ha^{-1}$) |
| 1–10 | 12 | 666.0 | 340.6 | 735.4 | 344.9 |
| 11–20 | 23 | 665.4 | 292.9 | 744.7 | 292.8 |
| 21–30 | 8 | 516.8 | 330.9 | 589.2 | 371.9 |
| 31–40 | 4 | 325.9 | 148.9 | 457.9 | 214.9 |
| 41–50 | 2 | 561.1 | 136.4 | 604.3 | 179.2 |
| 51–60 | 2 | 455.8 | 228.7 | 500.9 | 230.2 |
| 61–70 | 2 | 908.0 | 206.3 | 1177.9 | 255.0 |
| NHIER | 17 | 561.2 | 240.8 | 712.4 | 284.8 |

Note: NHIER—no human intervention ever registered.

Another confirmation that these forest ecosystems are capable of self-regulation is given by the decrease in structural differences between plots as the time elapsed since set-aside increased. Thus, the standard deviation in both the volume of living trees and the volume of dead wood successively decreased over time since forest management shifted to entire protection.

### 3.3.3. Number and Basal Area of Large Trees in Set-Aside Forests

In the first four decades since the set-aside, there is a progressive decrease in the number (Figure 8a) and percentage of large trees, but their number is much higher in the forest where the last harvest was registered more than 60 years ago. These figures depend on the characteristics of the forest stands at the time they were set aside and on the pace of ecosystem processes thereafter. The decrease in the number of large trees in the first decades is accompanied by an increase in the volume of dead wood (Figure 7). This decreasing trend also exists in relation to the basal area in the first 40 years after setting aside (Figure 8b), but the decreasing rate in the basal area percentage is higher than that of the number of trees. Thus, even though the dying trees are large, the effect on the basal area of all living trees is small (less than 10% in 40 years), compared to a 67% decrease in basal area if we consider only the large trees. This can be explained: (1) by the low percentage of basal area of large trees (8 to 22%) in the first four decades, compared to 40% registered in stands set aside more than 60 years ago; and (2) by the resilience of these forest ecosystems and the ability of remaining trees to valorize the new ecological conditions. Moreover, the number of large trees in stands where the last timber harvesting occurred more than 60 years ago is 1.8 times higher than the corresponding number in stands where no harvesting was ever performed, indicating that a long time is still necessary to entirely recover the characteristics of old-growth forests.

We noticed that higher harvesting intensities are associated with a decrease in the number of large trees (Figure 8c), even though the data are not sufficient to achieve statistical significance. In forest stands with no harvest activities in the past there are about 39 large trees per hectare (SDV = 34), while in stands where the harvested volume exceeded $6 \ m^3 \cdot yr^{-1} \cdot ha^{-1}$ in the recent past, no large tree was found.

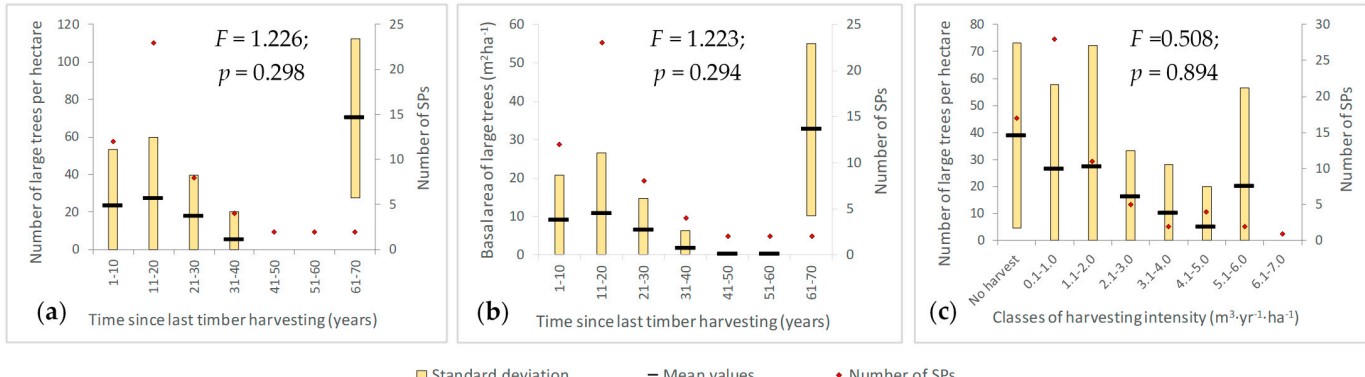

**Figure 8.** The number (**a**) and basal area (**b**) of large trees with respect to time since last harvest and harvesting intensity (**c**).

### 3.3.4. Natural Regeneration after Forests' Set-Aside

The natural regeneration process significantly depends on the time elapsed since the last forest operation ($F = 2.066$; $p = 0.0089$). In the stands where the latest wood harvesting is relatively recent, there are a large number of recruiting seedlings as a result of the gap management. In the case of stands where the last wood harvesting took place 51 to 70 years ago, the current number of recruiting seedlings is relatively small (about 3000 seedlings per hectare in average) (Figure 9), these stands having simple structures and low ages at the time the last forest operation occurred.

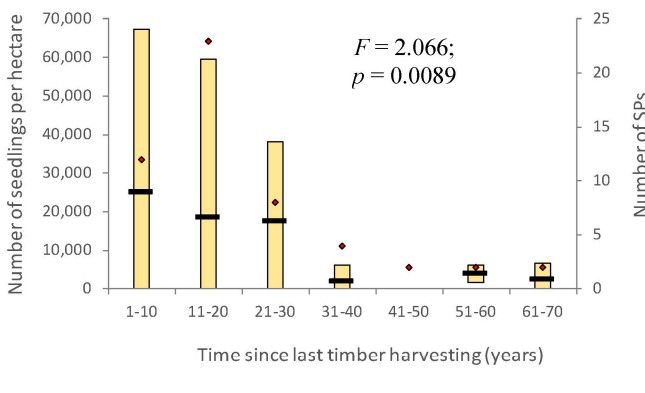

| Pairwise p-values | 1–10 | 11–20 | 21–30 | 31–40 | 41–50 | 51–60 | 61–70 |
|---|---|---|---|---|---|---|---|
| 1–10 | | 0.7038 | 0.0433 | 0.0314 | 0.0114 | 0.4422 | 0.2748 |
| 11–20 | 0.7038 | | 0.0826 | 0.0474 | 0.0106 | 0.3521 | 0.2792 |
| 21–30 | 0.0433 | 0.0826 | | 0.2811 | 0.3942 | 0.0905 | 0.5629 |
| 31–40 | 0.0314 | 0.0474 | 0.2811 | | 0.4709 | 0.2646 | 1 |
| 41–50 | 0.0114 | 0.0106 | 0.3942 | 0.4709 | | 0.343 | 1 |
| 51–60 | 0.4422 | 0.3521 | 0.0905 | 0.2646 | 0.343 | | 1 |
| 61–70 | 0.2748 | 0.2792 | 0.5629 | 1 | 1 | 1 | |

**Figure 9.** Tree species regeneration with respect to time period since last timber harvesting (green color highlights the significant differences: $p < 0.05$).

If we consider the time elapsed since set-aside, a slight decreasing tendency (statistically not significant) in the intensity of the natural regeneration process occurs once the entire protection period becomes longer. There is also a stabilization in the number of seedlings at about 7500 per hectare, a value very close to that identified for forest stands where no human intervention has ever been carried out. A more detailed analysis of the FMPs shows that in these stands, the latest forest operations consisted in harvesting accidental products or in sanitary cuttings, mainly focusing on harvesting dry trees and those felled for various reasons. These selective operations favored the opening of gaps in the stands, tree size diversification and the creation of favorable conditions for natural regeneration.

### 3.4. Set-Aside vs. Never-Harvested Forest Stands

We compared the set-aside forest stands with the "never-harvested" ones from N2000-RG, based on their structure type when set aside, on the one hand, and their current state, on the other hand (Figure 10). When we assessed the current state based on the number of large trees per hectare, we distinguished significant differences between stands ($F = 5.369$,

$p = 0.0011$), depending on their structure when set aside (Figure 10a). However, these differences are not significant between the uneven-aged (both, irregular ($p = 0.603$) and balanced ($p = 0.375$)) and the never-harvested stands, which may indicate that, at least partially, the former uneven-aged forest stands recovered their naturalness in terms of large trees numbers. The mean number of large trees per hectare is increasing from even-aged to uneven-aged balanced structure types, and a stabilization is expected at about 16 large trees per hectare, as found in never-harvested stands.

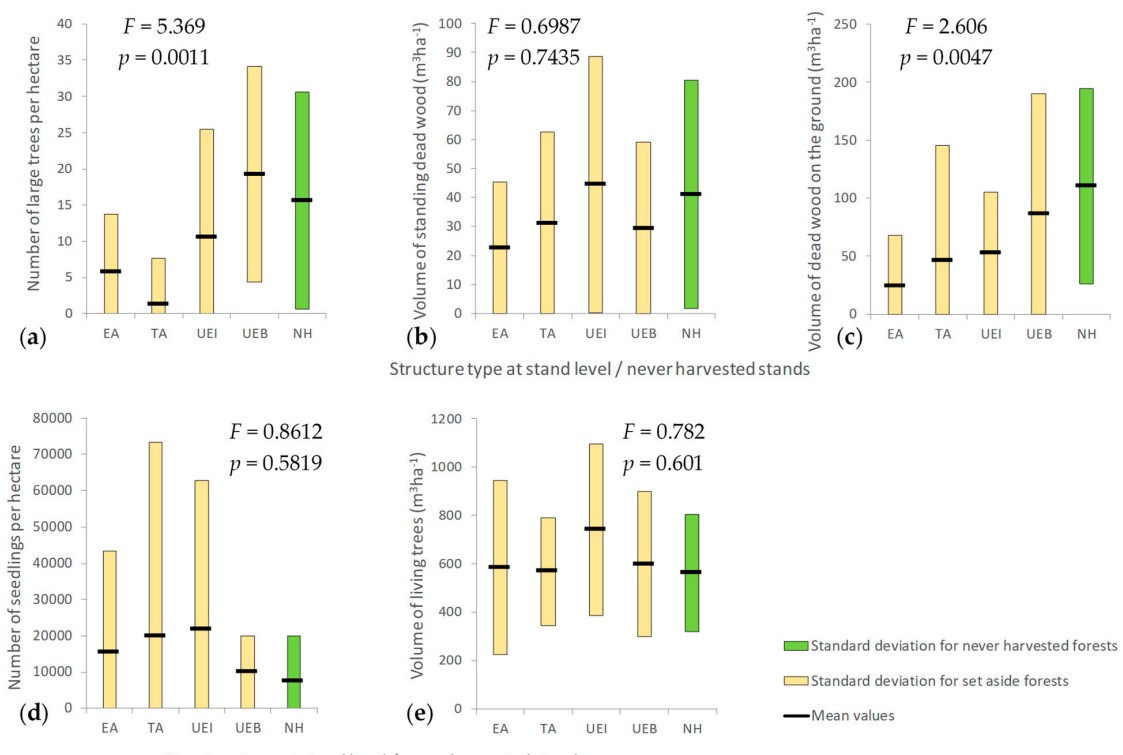

**Figure 10.** Closeness of set-aside forests to old-growth forests in terms of: (**a**) number of large trees per hectare, (**b**) volume of standing dead trees, (**c**) volume of deadwood on the ground, (**d**) natural regeneration, (**e**) volume of living trees (EA—even-aged; TA—two-aged; UEI—uneven-aged irregular; UEB—uneven-aged balanced; NH—never-harvested forest stands).

Significant differences were also found between the types of the set-aside forest structures when we analyzed the current volume of deadwood on the ground (Figure 10c). This volume is larger once the structure of the set-aside stand becomes more complex. The stands with uneven-aged balanced structures have best recovered the old-growth attributes in terms of deadwood on the ground, but there is still a difference of 21% that will probably decrease in the coming decades.

When we analyzed the volume of standing dead trees (Figure 10b), that of the living trees (Figure 10e) and the number of seedlings (Figure 10d) with respect to structure type when set aside, we found no significant differences between set-aside forests and the never-harvested ones, although some variation tendencies may be observed. For example, the uneven-aged irregular stands register the largest wood volumes of standing trees (both living and dead) and the most intense regeneration process, the mean number of seedlings per hectare being almost triple compared to never-harvested stands. The lack of significant differences between means in these cases is most likely a consequence of insufficient data in our analysis and does not indicate that the set-aside forests entirely recovered the old-growth characteristics. However, it can certainly be stated that the recovery process of the natural characteristics (as we found in old-growth forests) is much longer in the case of the even-aged, as compared to the uneven-aged stands.

## 4. Discussion

### 4.1. Structural Diversity with Respect to Past Forest Management

In different manners, all management types change some of the forest characteristics and it we are not likely to be able to assume that the type of forestry has no effect on biodiversity [68]. Our study highlights that structure diversity very much depends on past forest management and that the recovery process of old-growth characteristics is faster when we apply uneven-aged management with low-intensity cuts instead of even-aged management with high-intensity cuts. Assessment of forest structure diversity cannot neglect the tree size heterogeneity because, in the case of forest ecosystems, the biodiversity of all the other layers depends to a large extent on tree diameter structural diversity [69,70]. The dimensional heterogeneity of trees in set-aside forests can be higher than in managed (production) forests and also includes an old-growth phase, with very old and large trees [71]. In unmanaged stands, the size structure is shaped by spatial and temporal gap dynamics and neighborhood processes [69].

According to our data, there are stands where the last forest operation was recorded in the past 10–20 years with a high level of dimensional diversity, as well as stands in which the interventions ceased 41–60 years ago, but whose dimensional diversity is reduced. These contradictory findings of index values show that the tree size heterogeneity depends on the time elapsed since the last harvesting as well as on the main characteristics of the forests at the moment when they were set aside. Similar results were obtained by Pach et al. [69] who found that, in the case of the unmanaged forests, the Gini coefficient varied from 0.33 to 0.73, while Keren et al. [72] showed that the old-growth forests in Southeast Europe reach a mean value on the Gini index of 0.67. Additionally, the existence of extremely valuable stands in terms of tree size heterogeneity outside the strict protection areas from N2000-RG denotes a good forest management in the past, but also indicates the forest compartments that could be set aside in the near future in this protected area, as also recommended by Watson et al. [73].

### 4.2. The Variation Patterns of Structure Diversity Related Attributes—Driven by the Forest Stand Characteristics at the Moment When Set Aside

The structural characteristics of forest stands when set aside, especially stand structures, densities and ages, represent important drivers for the dynamic of deadwood, large trees numbers, the volume of living trees and natural regeneration. It is known that the presence and amount of deadwood in forest ecosystems are influenced by many factors such as above-ground living biomass, dominant tree species and the time elapsed since the last silvicultural intervention was applied [74]. Our findings have shown that the mean volume of standing dead wood registers the highest value in forest stands where no tree was ever harvested (38.7 $m^3 \cdot ha^{-1}$), while in the forests stands where low-intensity forest operations were applied, this volume is 7 to 30% lower. Other studies [42,75] present similar results, indicating that differences exist between managed and set-aside forests in terms of deadwood volume.

For the forests of Central and South-eastern countries, the dead wood volume varies between 30 and 290 $m^3 \cdot ha^{-1}$ [76], while in unmanaged beech forests it was $21.6 \pm 13.5$ $m^3 \cdot ha^{-1}$ [75]. In our study, the highest mean volume of deadwood (standing plus lying) (270 $m^3 \cdot ha^{-1}$) was found in forest stands where the last harvest occurred more than 60 years ago. This is in accordance with Nagel et al. [3], who indicated that, with respect to the management classes, the deadwood volume increases from an average of 15 $m^3 \cdot ha^{-1}$ to 165 $m^3 \cdot ha^{-1}$ and the volume of lying dead wood strongly increases after the last timber harvesting. In N2000-RG, in forest stands harvested more than 60 years ago, the mean volume of dead wood on the ground reached 263 $m^3 \cdot ha^{-1}$. For beech forest reserves in Central Europe, with a mean time span of non-intervention of 35 years, Vandekerkhove et al. [27] have put forward a mean value of $64.9 \pm 58.0$ $m^3 \cdot ha^{-1}$, which is in line with our findings for the same period of non-intervention.

The distribution of the dead wood volume by decay classes can be used to evaluate the time elapsed since the last forestry interventions [44,77]. Our results confirm this hypothesis, showing that the highest values of dead wood on the ground in the fifth decay class is found in stands where the last timber harvesting occurred 61 to 70 years ago. Similar results are presented by Nagel et al. [3], demonstrating that deadwood is present in all decay stages across the forest reserves, with a higher proportion of logs in advanced decay stages in the case of old-growth reserves, compared to young reserves.

The natural regeneration of trees depends on a large number of site factors: competing species, amount of dead wood, browsing, etc. [78–80], but also on the forest management type and its impact on these factors [81,82]. Our results show that, in case of mixed temperate forests from N2000-RG, the time elapsed since the last forest operation produces effects on the regeneration process. Within the plots from forest compartments recently set aside, due to the newly opened gaps as a result of the first fallen trees, the regeneration process is more active. After more than 30 years since the last harvest, we could not find a direct connection between natural regeneration and the other analyzed characteristics of these forests, because the entire N2000-RG tends to become homogenous in terms of forest structure, biodiversity, and live and dead wood volume.

### 4.3. Old-Growth Characteristics Not Entirely Recovered after 70 Years since Set-Aside

Our results, based on the Gini index for quantifying the tree size heterogeneity, prove that the index value in case of stands fully protected for almost 70 years is 24% higher than in the case of stands where the most recent harvest occurred in the past 10 years, and even exceeds the dimensional heterogeneity of stands from which timber has never been harvested. However, there are still significant differences in terms of the tree dimensional heterogeneity between the set-aside stands and those that have never been harvested. We thus can confirm that the process of recovering the old-growth structure is slow, as stated by Sabatini et al. [12].

The number and size of large trees are indicators of old-growth structures [15,16]. These trees are promoters of biodiversity and, in the medium to long term, the large old trees are a major source of deadwood. This influences the dynamics of dead wood in terms of availability and supply [77]. An important factor to consider is the time required for the number and size of large trees to recover after the abandonment of harvesting activities. According to our results, the number of thick trees tripled in forests last harvested more than 60 years ago, as compared to those where the most recent harvest occurred in the past decade. Moreover, the ratio between the number of large trees and the total number of living trees is 70% higher in the case of stands set aside 61 to 70 years ago, as compared to stands never affected by wood harvesting, and this is a consequence of the more diverse structure of old-growth forest stands, with the J-reversed shapes of trees' distribution in diameter classes, and thus a much higher number of living trees. The current lack of large trees in some set-aside forest stands is expected to be filled in the coming decades, particularly after large trees differentiation, as it is known that they develop large crowns and their competition for resources is low, which benefits favorable conditions for growth and development [83]. According to Larrieu et al. [23], the conservation of large trees should help to manage biodiversity in all forest ecosystems.

Another attribute related to old-growth forests is the deadwood volume in the fifth decay class which, in N2000-RG, is higher in the case of forests set aside 70 years ago than in the case of intact forest stands. Thus, we may infer that the set-aside forests have not been stabilized yet from the structural and functional perspective, although they have similar structures to those developed only under the impact of nature. As a consequence, the process of blurring the differences that still exist will continue in the coming decades. This is in accordance with other findings [15,27], which demonstrate that both the time of abandonment and management history are very important variables affecting deadwood availability in non-managed forests. The deadwood volume in set-aside forests will further

increase with the increasing time since abandonment, but it may be several decades before we have an important diversity of decay stages [24,44].

In the case of stands never affected by timber harvesting, we found that the deadwood volume represents about 26% of the living trees volume. Other findings [16] have shown that the ratios of coarse woody debris and live wood volume in Europe ranges from 6% to 89% in old-growth forests and from 2% to 9% in mature forests. In different forest types of the Austrian natural forest reserves, this ratio is 12% and tends to increase once the silvicultural interventions are reduced [84]. Thus, the ratio becomes an important indicator of a forest system maturation under the prevailing drive of natural factors.

## 5. Conclusions

Once we know that forests will be seriously affected by future disturbances, the study of forest ecosystems never affected by human activities, or not affected any more due to the abandonment of harvesting practices, becomes extremely important for a holistic understanding of ongoing natural processes, in order to implement such knowledge in current and future forest management. To adapt future forest management to new social, ecological and economic conditions, it is important to know the former management and to learn from previous experiences. Former management modifies forest functioning and, consequently, tree size heterogeneity. We observed that the recovery of old-growth characteristics after set-aside greatly differs, depending especially on management type and stands' characteristics at the time of exclusion from regular management. The past management type and harvesting intensity on the one hand, and stands structures, densities, and ages, on the other hand, are the main factors that influence the rhythm of naturalness recovery.

The obtained results allowed us to highlight the way in which the temperate forest ecosystems in mountain areas react after the exclusion of the human factor. Thus, it was found that the first 40 years after the last timber harvest are characterized by a structural reshape of stands, describing a transition period with low resilience forests, increased tree mortality, a significant decrease in the regeneration processes intensity, and a reduction in the total number of trees (and especially the number of large trees), all these aspects leading to a decrease in tree size heterogeneity. After these four decades, the values of the mentioned parameters have improved and, after 60 to 70 years since the last harvest, the ecosystem processes and descriptors have started to become similar to those in the forests unaffected by human actions. However, our results demonstrate that 70 years of no timber harvesting are not sufficient for a complete recovery of old-growth characteristics.

**Supplementary Materials:** The following supporting information can be downloaded at: https://www.mdpi.com/article/10.3390/f14020251/s1, Table S1. The analysed FMPs from management units which previously overlapped on the current area of N2000-RG, Table S2. Coefficients used for computing the tree above-ground wood volume, Table S3. Short overview of the history of protected forest areas from the reserves inside N2000-RG. References [49–51,54,65,85–105] are cited in the supplementary materials.

**Author Contributions:** Conceptualization, G.D.; Data curation, C.-O.B., G.D. and C.V.T.; Formal analysis, C.-O.B., G.D. and C.V.T.; Funding acquisition, G.D.; Investigation, C.-O.B. and G.D.; Methodology, C.-O.B., G.D. and C.V.T.; Project administration, G.D.; Resources, G.D. and C.-O.B.; Software, G.D.; Supervision, C.-O.B. and C.V.T.; Validation, C.-O.B., G.D. and C.V.T.; Visualization, G.D.; Writing—original draft, C.-O.B. and G.D.; Writing—review and editing, C.-O.B., G.D. and C.V.T. All authors have read and agreed to the published version of the manuscript.

**Funding:** Data collection within this research was funded through the contract number 9784/2014. The publishing fee was assured through the Romanian Ministry of Research, Innovation and Digitalization within Program 1—Development of national research and development system, Subprogram 1.2—Institutional Performance—RDI excellence funding projects, under contract no. 10PFE/2021.

**Data Availability Statement:** Online data sources and software platforms World Imagery Sources: Esri, DigitalGlobe, GeoEye, i-cubed, USDA FSA, USGS, AEX, Getmapping, Aerogrid, IGN, IGP, swisstopo, and the GIS User Community.

**Acknowledgments:** We are grateful to the responsible staff from the Suceava Forest Directorate, Crucea Forest District, Pojorâta Forest District and Stulpicani Forest District for providing access to forest management plans. Our appreciation goes also to Barnoaiea Ionuț from "Stefan cel Mare" University of Suceava for the GIS support. Four anonymous referees provided insightful comments on this manuscript. Pășcuț Gheorghe-Lucian and Ghețu Cornelia Luminița kindly corrected the English errors.

**Conflicts of Interest:** The authors declare no conflict of interest.

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
