# Peer review of "Variation Patterns of Forest Structure Diversity after Set-Aside in Rarău-Giumalău Mountains, Romania"

_forests, doi:10.3390/f14020251_

Round 1

Reviewer 1 Report (New Reviewer)

The issue addressed in this article, related to the return of managed forests to "natural" forest structures following time since abandonment, is currently very important to guide conservation/restoration policies in Europe. The authors seem to have access to relevant data to contribute effectively to this topic. In its current version, however, the article has to my opinion important methodological weaknesses that strongly impact the clarity or robustness of the results. For this reason, the article cannot be accepted for publication in its current form. My reserves concern in particular five points:

- The authors separate the factors of abandonment (time since last cut) and protection (time since set-aside) in most of the paper, but this causes an important redundancy in the results. What is the interrelationship of these values and is there any real value in keeping them separate and analyzing them both? Here the authors can look at the methodologies of articles they cite (e.g., Thompson et al. 2022) to adjust this point.

- There is a significant bias regarding the time-since-abandonment, i.e., what type of harvesting was last done and in what forest type was not considered. The effects of a harvest vary greatly between a clear cut, a thinning or a selection cut in stands of different ages. For this reason, time-since-abandonment cannot be studied without also considering the possible “noise” caused by the grouping of stands with very different histories (something that can be seen through the very high standard-deviations in the Figures). If I refer to line 242, this data seems to be available. It is therefore necessary to take it into consideration in the analyses.

- The authors consider a broad set of forest structure metrics in their analyses but they are all analyzed separately by means of Kruksal-Wallis analysis. This makes the results complex and redundant, making it difficult to draw a clear result from this study. It is also possible that many of these variables are highly intercorrelated, reinforcing this redundancy. The use of a multivariate approach coupled with ANOSIM or PERMANOVA type analyses, for example, could thus clearly identify the overall trends that forest abandonment/protection have

- In lines 243-246, the authors separate different types of structure to which they assign an arbitrary numerical value (1-4) and then make a numerical comparison of these values (Table 2). This approach cannot be accepted in a peer-reviewed scientific article: this is categorical data and should be treated as such. There is no relevant reasoning to justify such a choice: a one-sized balanced structure is not worth 4 times as much as an even-sized structure and 2 times as much as a two-sized structure, this does not make sense. This analysis needs to be revised.

- The article uses too many abbreviations, often unjustified. For example, why talk about "sample plots" and then use the acronym "SP" when "plots" would be just as clear? Some abbreviations are also used before they are presented, for example "FMP" which is only presented on line 230 when it is already used three times. This clouds the clarity of the article.

For these reasons, I believe that a major reworking of the method and results (and consequently partly of the discussion) is necessary before this article can be accepted for publication. The deadlines assigned by Forests for major revisions, however, are often too short to allow authors to revise their article in proper conditions. This would be unfortunate because this research has real potential for the scientific community and I would be pleased to be able to read a revised version of the article. I would therefore advise authors to take their time to work on their revision of the article, even if it has to go far beyond the deadlines requested by the journal.

Author Response

Dear Reviewer,

Thank you very much for your comments!

We addressed every comment, attempting to provide clarification. We have considered constructively each observation, but we believe that certain aspects have not been well understood. We hope that our responses and changes to the text (where applicable) are beneficial and have contributed to the improvement of the manuscript.

Reviewer comment

Authors response

1.

The authors separate the factors of abandonment (time since last cut) and protection (time since set-aside) in most of the paper, but this causes an important redundancy in the results. What is the interrelationship of these values and is there any real value in keeping them separate and analyzing them both? Here the authors can look at the methodologies of articles they cite (e.g., Thompson et al. 2022) to adjust this point.

The analyzed site is characterized by certain particularities regarding the management history. They have been described in the article, but we bring some clarifications to eliminate any confusion.

In respect to this first comment, we consider the analysis should be maintained as done before because, at least for the studied area, there are important differences between „time since set aside” and „time since last cut”. Thus, there are forest stands for which the time since the last cut is much longer than the time since set-aside. Also, as shown in the article (section 2.2), in the case of the Giumalău forest reserve, „part of the reserve was transformed in buffer area from 1983 to 1994, due to the damages produced by wind in some forest compartments and to that time decision to extract the damaged wood” (also presented in figure 2.a). Thus, in case of some plots, even they were previously part of set aside forest areas, trees were harvested and, after that, the stands were re-included in strictly protection regime.

Therefore, we consider the differences between „time since set aside” and „time since last cut” are clearly presented in the paper and, consequently, no additional changes were done.

Furthermore, we know well the article published by Thompson et al. 2022, which we consider extremely useful, but the methodology they propose cannot be applied ad-litteram everywhere. In our case the research network overlaps a smaller area, only public forests, located only in mountains, being more homogenous, with only three dominant tree species: Norway spruce, silver fir and European beech. As Thompson et al. 2022 did, we also excluded some plots in different steps of our analysis (in the beginning we kept in the analysis 319 plots from a total of 347 and later we only conducted our analysis considering the 70 plots from 1-PRP), but considering the methodological aspects which were important to address our research objectives.

2.

There is a significant bias regarding the time-since-abandonment, i.e., what type of harvesting was last done and in what forest type was not considered. The effects of a harvest vary greatly between a clear cut, a thinning or a selection cut in stands of different ages. For this reason, time-since-abandonment cannot be studied without also considering the possible “noise” caused by the grouping of stands with very different histories (something that can be seen through the very high standard-deviations in the Figures). If I refer to line 242, this data seems to be available. It is therefore necessary to take it into consideration in the analyses.

We agree with you to some point. A long time analysis is difficult to achieve without successive inventories. In the present case, the analysis is based on the data collected at a first forest inventory (and the only one for the moment) carried out in the installed permanent network. As mentioned in the manuscript, our analysis was based on data collected from 319 inventory plots, out of which 249 are in the 2-PRP, installed in stands from which trees have never been harvested and which were in strictly protection during the entire period analysed in this study.

Thus, the influence of the management type on forest diversity was studied primarily based on the information collected from the SPs included in the strictly protection regime after 1941 (especially SPs from 1-PRP). In the same time, the areas from 2-PRP constituted a solid data source for describing the target natural structure the cultivated stands in the area tend for.

As indicated in the manuscript, the 1-PRP overlaps a total area of about 2070 ha of forest. These forests are divided into 332 compartments. This leads to an average ratio of 1 SP per 5 forest compartment. Considering that the average area of the compartment is 6.23 ha, and the total inventoried area corresponding to the 70 SPs is 3.5 ha, a detailed analysis as you asked would be irrelevant because it is impossible to determine the exact location of the cuttings carried out in the studied stands in various periods of time.

We extracted from management plans data regarding the year and type of last forest operation but, as mentioned in section 2.2 of the manuscript, in the studied area, in the stands over which the plots overlap, only low intensity and selective cuttings were planned and carried out during the analyzed period (1941-2015) (so, no clear cuttings). Unfortunately, information on the precise location of the cuttings inside the forest compartments does not exist in the forest districts archives, so we could not connect them directly to our sample plots.

For these reasons the analysis of past forest management was done only in respect to "time since last harvesting" and "time since set aside" (both at compartment level). These are the only certain information regarding the forest management activities in the compartments and, only based on them, we identified the "very different histories" you mentioned.

Considering your comment, we added the following sentences in lines 247-252:

„Our analysis was based on data collected at forest compartment level for the entire studied period, as the FMPs are the only sources for historical data related to forest management in the studied area. Information on the precise location of the cuttings inside the forest compartments does not exist in the forest districts archives, so these historical data were indirectly connected to our sample plots, based on spatial locations of SPs and forest compartments”.

3.

The authors consider a broad set of forest structure metrics in their analyses but they are all analyzed separately by means of Kruksal-Wallis analysis. This makes the results complex and redundant, making it difficult to draw a clear result from this study. It is also possible that many of these variables are highly intercorrelated, reinforcing this redundancy. The use of a multivariate approach coupled with ANOSIM or PERMANOVA type analyses, for example, could thus clearly identify the overall trends that forest abandonment/protection have.

We agree that a statistical analysis of these data is difficult to achieve, especially since different specialists propose different statistical methods. ANOSIM and PERMANOVA are methods similar to those already applied in the article for checking the null hypothesis and analysing the differences between classes. Therefore, we consider that a new statistical analysis and, consequently, rewriting the entire article are not necessary, since the results obtained will be similar to the current ones. More than that, the statistical analysis we used is commonly used for this type of studies.

4.

In lines 243-246, the authors separate different types of structure to which they assign an arbitrary numerical value (1-4) and then make a numerical comparison of these values (Table 2). This approach cannot be accepted in a peer-reviewed scientific article: this is categorical data and should be treated as such. There is no relevant reasoning to justify such a choice: a one-sized balanced structure is not worth 4 times as much as an even-sized structure and 2 times as much as a two-sized structure, this does not make sense. This analysis needs to be revised.

Partially we agree with you. This type of structural index is not presented and used in literature. However, in forest management planning in Romania, the type of structure is established at the stand level in the same way since the beginning of the 20th century, and the use of these numerical values assigned to the four types of structure is clearly regulated since 1948, through the Romanian Technical Norms for forest management planning. So it is a clear information, which we find it in all analyzed FMPs, for each forest stand, just as we find information regarding age, stand density, composition, etc.

We didn’t want to replace or assimilate it with any other index recommended by the literature for quantifying the structural diversity of stands in order not to distort historical information. For this reason, we consider that, for the present analysis, the approach is acceptable, given that there is no other continuous information in the FMPs that would allow the calculation of an index able to ensure the comparability of the information over the entire studied period.

5.

The article uses too many abbreviations, often unjustified. For example, why talk about "sample plots" and then use the acronym "SP" when "plots" would be just as clear? Some abbreviations are also used before they are presented, for example "FMP" which is only presented on line 230 when it is already used three times. This clouds the clarity of the article.

Thank you very much for this observation! We have revised the entire text of the article and used acronyms only where absolutely necessary (e.g. tables, figures).

We explained the meaning of FMP abbreviation in line 181, when we firstly used the term.

Reviewer 2 Report (New Reviewer)

In my opinion the paper is very well prepared and written and provides very important data. I consider that it can be published in its current form. I found only few lines where grammer should be carefully checked:

Line 22, "begins to increase"

Line 45, "nature-based"

Perhaps not all formulas are needed, especially No.4, which is clearly described in the text above

The title of Table 2, I was confused understanding it. Is all the data related to the Gini index?

If there should be some revision the Discussion can be shortened as the data is very well described and the conclusions are really solid.

Author Response

Dear Reviewer,

Thank you very much for your observations!

We addressed every comment, attempting to provide clarification. We hope that our responses and changes to the text (where applicable) are beneficial and have contributed to the improvement of the manuscript.

Reviewer comment

Authors response

1.

Line 22, "begins to increase"

We believe that your comment refers to the sentence section „ The volume of lying dead wood increased …” and probably you suggest to change „increased” with „begins to increase”. However, in the same line, the previous sentence ends with „… the diversity begins to increase”. We tried to avoid repeating the same wording in the same line and, in order to maintain the initial meaning, we kept the text as it was in the initial manuscript.

2.

Line 45, "nature-based"

We changed „nature base” to „nature-based”.

3.

Perhaps not all formulas are needed, especially No.4, which is clearly described in the text above.

We have included in section 2.3.2 only those formulas strictly necessary for understanding the main methodological steps taken in data processing. For this reason, to ensure easy understanding of the methodology, we consider formula 4 should be maintained in the paper.

4.

The title of Table 2, I was confused understanding it. Is all the data related to the Gini index?

Table 2 presents the mean values of stands characteristics in the moment they included in entirely protection regime. We needed these historical data about stand characteristics in order to support mainly the interpretation of dimensional trees heterogeneity assessed by using the Gini index (section 3.1), but also of the dead wood (L 356-358 and 384-389) and of natural regeneration (L 456-458) (lines numbering corresponds to initial form of the manuscript).

5.

If there should be some revision the Discussion can be shortened as the data is very well described and the conclusions are really solid.

Thank you for your suggestion! We reanalyzed the Discussion section, and we consider that shortening it might affect the fluency of the paper. Therefore, we kept it in the initial form.

Reviewer 3 Report (New Reviewer)

Authors presented a very nice dataset, which is original and can be the source of very interesting results. However, my main concern is that they didn't extracted the substantial information from it, mainly by not asking accurate questions or stating hypotheses.

To me, this work need to be published but it first need major revision and thus a significant work to be done.

General comments:

Generally, from the title to the conclusion, the word « diversity » is misused. Forest diversity refer to composition what was not studied here. You measured and compared forest structure and found a diversity of it, but please to avoid misleading the reader, refer to forest structure diversity or maybe forest heterogeneity.

You lack good hypotheses. You produced and now possess an incredible interesting dataset and you only proposed 2 objectives. The first one being methodological, you only have a single and vague objective that guides the discussion towards a series of confirmations and very few statements. Why not stating some hypotheses here? For instance: Based on the time needed for naturel functions to recover, we expect that after 80 years set aside, forests will not express similar characteristics and NHIER forests? You can also add « depending on management type before set aside, forests will recover faster the NHIER characteristics ». This lack of hypothesis testing leads to a rather confirmatory discussion that is not up to par.

You need to (and to me, you have to) describe better the management that occurred on the different places. You should have that information with your study of past activities. This is critical since forest recovery will be very different from even-aged or uneven-aged management. As suggested above, you can even include that in your hypothesis and verify that recovery after even-aged management should be longer than after uneven-aged management.

To me you should not show both graphs (Time since last harvest and Time since protected). Choose on and put the other one in supplementary material. Time since last harvest seems to be more ecologically based...

More specific comments:

Introduction:

I don’t really see the thread of the introduction. The structure of the introduction appears like a list of (true but scarce) information that leads with difficulty the reader to your objective and questions.

Your second objective refer to diversity which is not the case. Your are studying forest structure not diversity.

Finally, why not looking at past forest management intensity? This is critical in the way and the time needed for a managed forest to recover natural old-growth characteristics.

L42, impact on biodiversity comes first and then impact on services arises.

L109, how can forest management types can affect forests that are set-aside? Are talking about previous forest management?

Results:

Why presenting fig3, and 4 with all or only 1-PRP species? Results seem similar.

In fig 6, 7 and 8 please simplify by presenting only one approach: time since protection or Time since last harvest (the latter being more ecologically based.

Discussion:

Globally, it could be reduced by 20%

4.2 All this is very confirmatory. From 527 to 559 you can be much more concise.

4.3 and 4.4 are better but could be slightly reduced.

Conclusion: 

I don’t like the word « behavior » for forest ecosystems but even if you are right that their differences came from their characteristics at time of protection, it would have be much more interesting to explain why: Are these

different characteristics coming from contrasting harvest intensity (even- or uneven-aged management) or time since last harvest, or station fertility…?

The second part of the conclusion is much more interesting but stay with time since last harvest. Set aside is a political decision that has nothing to do with ecology.

It seems to me that the English should be reviewed. 

Literature of interest:

Roy, M. È., Surget-Groba, Y., Delagrange, S., & Rivest, D. (2021). Legacies of forest harvesting on soil properties along a chronosequence in a hardwood temperate forest. Forest Ecology and Management, 496, 119437. doi: 10.1016/J.FORECO.2021.119437

Watson, J. E. M., Evans, T., Venter, O., Williams, B., Tulloch, A., Stewart, C., … Lindenmayer, D. (2018). The exceptional value of intact forest ecosystems. Nature Ecology & Evolution 2018 2:4, 2(4), 599–610. doi: 10.1038/s41559-018-0490-x

Author Response

Dear Reviewer,

Thank you very much for your recommendations and encouragements!

We addressed every comment and we hope that our responses and changes to the text are beneficial and have contributed to the improvement of the manuscript.

Reviewer comment

Authors response

1.

Generally, from the title to the conclusion, the word « diversity » is misused. Forest diversity refer to composition what was not studied here. You measured and compared forest structure and found a diversity of it, but please to avoid misleading the reader, refer to forest structure diversity or maybe forest heterogeneity

We revised the entire paper considering your recommendation. The title was adjusted accordingly.

2.

You lack good hypotheses. You produced and now possess an incredible interesting dataset and you only proposed 2 objectives.

The first one being methodological, you only have a single and vague objective that guides the discussion towards a series of confirmations and very few statements.

Why not stating some hypotheses here? For instance: Based on the time needed for naturel functions to recover, we expect that after 80 years set aside, forests will not express similar characteristics and NHIER forests? You can also add « depending on management type before set aside, forests will recover faster the NHIER characteristics ». This lack of hypothesis testing leads to a rather confirmatory discussion that is not up to par.

Thank you very much for this remark! We reorganized the introduction, included work hypotheses and rewrote the research objectives.

3.

You need to (and to me, you have to) describe better the management that occurred on the different places. You should have that information with your study of past activities. This is critical since forest recovery will be very different from even-aged or uneven-aged management. As suggested above, you can even include that in your hypothesis and verify that recovery after even-aged management should be longer than after uneven-aged management.

We detailed at the end of Section 2.2 of the manuscript the silvicultural operations applied in the study area. We considered your recommendation and included a new subchapter in Results section (3.4), explaining that the recovery process is longer in case of uneven-aged compared to eve-aged stands.

4.

To me you should not show both graphs (Time since last harvest and Time since protected). Choose on and put the other one in supplementary material. Time since last harvest seems to be more ecologically based...

We only kept in our analysis the graphs referring to time since last harvest.

5.

I don’t really see the thread of the introduction. The structure of the introduction appears like a list of (true but scarce) information that leads with difficulty the reader to your objective and questions.

We revised the Introduction in order to assure the fluency of the information.

6.

Your second objective refer to diversity which is not the case. You are studying forest structure not diversity.

We reformulated the objectives according to the preliminary research hypotheses.

7.

Finally, why not looking at past forest management intensity? This is critical in the way and the time needed for a managed forest to recover natural old-growth characteristics.

Thank you for this suggestion. We included it in our analysis. The Method, Results and Discussion sections were adjusted accordingly.

8.

L42, impact on biodiversity comes first and then impact on services arises.

Agree, we reformulated:

„ It is well known that forests offer a wide range of ecosystem services, and the increasing anthropogenic pressure on them has a direct impact on biodiversity and, thus, on the provision of these services.”

9.

L109, how forest management types can affect forests that are set-aside? Are talking about previous forest management?

Indeed, we are talking about the previous forest management. We corrected the phrase:

„… to figure out how various management types applied in the past may affect biodiversity …”

10.

Results:

Why presenting fig3, and 4 with all or only 1-PRP species? Results seem similar. In fig 6, 7 and 8 please simplify by presenting only one approach: time since protection or Time since last harvest (the latter being more ecologically based.

We agree with your comment. We kept only the results from 1-PRP and the second approach: time since last harvest. We adjusted accordingly the entire manuscript.

11.

Discussion:

Globally, it could be reduced by 20%.

4.2 All this is very confirmatory. From 527 to 559 you can be much more concise.

4.3 and 4.4 are better but could be slightly reduced.

We entirely revised the Discussion section.

12.

Conclusion:

I don’t like the word « behavior » for forest ecosystems but even if you are right that their differences came from their characteristics at time of protection, it would have be much more interesting to explain why: Are these different characteristics coming from contrasting harvest intensity (even- or uneven-aged management) or time since last harvest, or station fertility…? The second part of the conclusion is much more interesting but stay with time since last harvest. Set aside is a political decision that has nothing to do with ecology.

We reformulated the phrase and the word «behaviour» was not used anymore. We revised the Conclusions, focusing better on our findings, explaining where these differences came from and considering the «time since last harvest».

13.

It seems to me that the English should be reviewed.

We also revised the English.

14.

Literature of interest:

Roy, M. È., Surget-Groba, Y., Delagrange, S., & Rivest, D. (2021). Legacies of forest harvesting on soil properties along a chronosequence in a hardwood temperate forest. Forest Ecology and Management, 496, 119437. doi: 10.1016/J.FORECO.2021.119437

Watson, J. E. M., Evans, T., Venter, O., Williams, B., Tulloch, A., Stewart, C., … Lindenmayer, D. (2018). The exceptional value of intact forest ecosystems. Nature Ecology & Evolution 2018 2:4, 2(4), 599–610. doi: 10.1038/s41559-018-0490-x

We considered tour recommendation.

Reviewer 4 Report (New Reviewer)

Title is quite ambitious. It need to be more specific. I suggest to add at the end “in Rarău-Giumalău (Romania)”.

Abstract must be improved.

Move the objective after the introduction sentences, and before the methods. Besides, the results are pure descriptive, add some statistics. The conclusion must be derived from your analyses. Please, check it.

In line 69, you describe silviculture. This is quite broad: intermediate treatments? Regeneration treatments? Please, add more information about this point.

Fig 2 quality must be improved. Captions must be bigger. Remove the horizontal lines.

Most of the formulas are not necessary. Please remove them. They can be described in the text. This is a scientific paper, not a scholar book text.

Fig 3 to 8. The quality and the cap size must be improved. I suggest to improve them. You can also use colour in the bars.

Conclusions are not conclusions; they are a brief summary of the paper. The conclusions must be short, and just highlight the main findings of the paper. The new knowledge for the science. You must better describe and synthetize your findings.

Author Response

Dear Reviewer,

Thank you very much for your comments!

We addressed every comment, attempting to provide clarification. We have considered constructively each observation, and we hope that our responses and changes to the text are beneficial and have contributed to the improvement of the manuscript.

Reviewer comment

Authors response

1.

Title is quite ambitious. It need to be more specific. I suggest to add at the end “in Rarău-Giumalău (Romania)”.

We considered your suggestion and change the title accordingly:

„Variation patterns of forest structure diversity after set-aside in Rarău-Giumalău mountains, Romania”

2.

Abstract must be improved.

We revised the Abstract, based on the changes we carried out in the manuscript.

3.

Move the objective after the introduction sentences, and before the methods. Besides, the results are pure descriptive, add some statistics. The conclusion must be derived from your analyses. Please, check it.

Considering also the observations of the other reviewers, we entirely reformulated the objectives and rewrote the Results section, including also the statistical analysis based mainly on Permanova test.

The objectives are stated at the end of Introduction section, before the Materials and Methods section.

Conclusion section was revised.

4.

In line 69, you describe silviculture. This is quite broad: intermediate treatments? Regeneration treatments? Please, add more information about this point.

In the Introduction section we only provided a brief description on the effect the silvicultural treatments have on forests diversity. Of course, the silvicultural treatments refers to regeneration cuts. We believe this is a good observation and we detailed at the end of Section 2.2 of the manuscript the silvicultural operations applied in the study area.

5.

Fig 2 quality must be improved. Captions must be bigger. Remove the horizontal lines.

We improved the quality of Figure 2, considering your recommendations.

6.

Most of the formulas are not necessary. Please remove them. They can be described in the text. This is a scientific paper, not a scholar book text.

We deleted the unnecessary formulas, and only kept their short descriptions.

7.

Fig 3 to 8. The quality and the cap size must be improved. I suggest to improve them. You can also use colour in the bars.

We modified all the graphs, taking into consideration your suggestion.

8.

Conclusions are not conclusions; they are a brief summary of the paper. The conclusions must be short, and just highlight the main findings of the paper. The new knowledge for the science. You must better describe and synthetize your findings.

We revised the Conclusions, focusing better on our findings.

Round 2

Reviewer 1 Report (New Reviewer)

I have carefully read the corrections as well as the answers provided by the authors, and I must express my surprise. I agree that the revision process is partly an exchange and a matter of compromise. The authors are not obliged to take into account all the requests made by the reviewers, as long as relevant arguments are presented. At the same time, it is necessary to take a minimum into account the opinion of the reviewers.

In the case of this article, the authors have made only the most minor corrections and have rejected all major points, usually with unconvincing arguments. For example, the accumulation of Kruksall-Wallis tests for a large number of variables, often very close together, is always to the detriment of the clarity of the article, no matter how often this test is used in other studies. Similarly, transforming structures into numerical variables to compare them with each other is still unacceptable for a scientific publication in peer-reviewed journals, regardless of whether it is used in the grey literature.

Given the results of this first round of review, it seems unnecessary to continue the review process. I apologize to the authors, but I must therefore recommend a rejection of the article. I cannot support its publication in its present form and it does not seem destined to change.

Author Response

Dear Reviewer,

We addressed your comments and we believe that the current form of the paper in very much improved compared with the previous one.

Thank you very much for your recommendations!

Reviewer comment

Authors response

1.

The accumulation of Kruksall-Wallis tests for a large number of variables,often very close together, is always to the detriment of the clarity of the article, no matter how often this test is used in other studies.

We modified the statistical analysis and used PERMANOVA instead of Kruskall-Wallis tests. The Results section was entirely modified, considering your observation.

2.

Transforming structures into numerical variables to compare them with each other is still unacceptable for a scientific publication in peer-reviewed journals, regardless of whether it is used in the grey literature.

We replaced the numerical variables used for structure stand description with the usual names of forest structures and replaced Table 2 with Figure 6.

This manuscript is a resubmission of an earlier submission. The following is a list of the peer review reports and author responses from that submission.

Round 1

Reviewer 1 Report

The authors tried to analyze the impact of previous forest management on current forest tree biodiversity. The variation trend of several indices that the forests diversity was described in relation to, which include trees size heterogeneity; wood volumes homogeneity of the living trees throughout the site; variability of the standing and lying dead wood volume; number and basal area of large trees; natural regeneration, were depicted in this manuscript. These results can promote the understanding of the impacts of forest management strategy on forest tree diversity. The manuscript matches the scope of the journal Forests.

However, I have some concerns on this manuscript:

1.     The biggest concern I have on this manuscript is that the analysis and discussion on the influence of the past management on the forest diversity is too weak and even missing. Firstly, in the results, the authors described the variation trends of diversity-related forest characteristics, while little analysis on the mechanism-relationships between the past management and forest diversity was done and shown in this manuscript. Moreover, in the discussion, the authors mainly compared result of the current study to those of the previous studies, while I did not see any contents on the discussion of the mechanism of the influence of the past management on the forest diversity. This does not match the current title of this manuscript that “past forest management shapes the trees diversity in „set-aside” forests”. So that, if there is no more analysis, results and discussions on the mechanism of the influence of the past management on the forest diversity, the manuscript title should be “variation patterns of tree diversity-related forest characteristics after set-aside” rather than the current one.

2.     The abstract, introduction and conclusion sections should be condensed.

3.     The section 3.1 in lines 234-265 is the introduction of the study areas. It is better to be moved to the materials and methods section.

4.     In sections 3.2-3.6 in lines 266-434, the authors just show the variation trends of diversity-related forest characteristics. Therefore, the words, such as, ‘impact’, ‘influence’, and ‘as a result’, are better to be deleted.

5.     The first two paragraphs of the discussion section in lines 436-451 is introduction to the background of this study rather than discussion. So, it is better to be moved to the introduction section.

6.     In the discussion, there is so many contents that repeated the results. For example, lines 455-470, 481-484, etc. The authors should check carefully and reorganize these sentences and paragraphs.

Author Response

Dear Reviewer,

Thank you very much for your constructive comments. We addressed every comment with appropriate changes in the paper and we justified below the changes.

Reviewer comment

Authors response

1.

The biggest concern I have on this manuscript is that the analysis and discussion on the influence of the past management on the forest diversity is too weak and even missing.

Firstly, in the results, the authors described the variation trends of diversity-related forest characteristics, while little analysis on the mechanism-relationships between the past management and forest diversity was done and shown in this manuscript. Moreover, in the discussion, the authors mainly compared result of the current study to those of the previous studies, while I did not see any contents on the discussion of the mechanism of the influence of the past management on the forest diversity.

This does not match the current title of this manuscript that “past forest management shapes the trees diversity in „set-aside” forests”. So that, if there is no more analysis, results and discussions on the mechanism of the influence of the past management on the forest diversity, the manuscript title should be “variation patterns of tree diversity-related forest characteristics after set-aside” rather than the current one.

We agree with your observation. Unfortunately, at the moment, we only have one census of the SPs in the study area and, indeed, a detailed analysis at SP level on the mechanism-relationships between the past management and forest diversity is difficult based only on this single census. We had to refer to the existing historical information in the FMPs, so to data at the level of the forest compartment, not at the level of SP. For these reasons, we agree with the suggestion to change the title. The new title of the article is the one you proposed: “Variation patterns of tree diversity-related forest characteristics after set-aside”.

2.

The abstract, introduction and conclusion sections should be condensed.

The „Abstract„ and the „Introduction” sections were condensed and the „Discussions” section was improved according to both reviewers’ suggestions. We kept the „Conclusion” section as it was in previous form of the manuscript because, in our opinion, it should not be condensed, having only two concise paragraphs.

3.

The section 3.1 in lines 234-265 is the introduction of the study areas. It is better to be moved to the materials and methods section.

Section 3.1 was moved to Material and methods as Section 2.2.

4.

In sections 3.2-3.6 in lines 266-434, the authors just show the variation trends of diversity-related forest characteristics. Therefore, the words, such as, ‘impact’, ‘influence’, and ‘as a result’, are better to be deleted.

We considered the remark and made appropriate changes in the „Results” section of the manuscript.

5.

The first two paragraphs of the discussion section in lines 436-451 is introduction to the background of this study rather than discussion. So, it is better to be moved to the introduction section.

These paragraphs were moved to „Introduction”.

6.

In the discussion, there is so many contents that repeated the results. For example, lines 455-470, 481-484, etc. The authors should check carefully and reorganize these sentences and paragraphs.

We revised the „Discussion” section, keeping only the most important figures for comparison with those from similar studies and trying to avoid being repetitive with the results.

Reviewer 2 Report

Review of “Past forest management shapes the trees diversity in“ set-aside” forests” by Cătălina-Oana Barbu, Gabriel Duduman and Cezar Valentin Tomescu

The manuscript aims to analyze the set-aside impact on the current diversity of mountain temperate forests from the Natura 2000 of the Rumanian site Rarău-Giumalău. The dataset is important since they analyze some sites with no past evidence of human intervention and several other sites where timber harvesting was done several decades ago. The main objective is to study the effect of time elapsed from last human impact (or time from strict reserve establishment) on structural diversity (measured by Gini index on basal area), on the amount of dead wood still in the forest, on large tree density and on regeneration. The general expectation is that the more time has passed since last human intervention, the more diverse should be the structural diversity of the site and more dead wood, the more large trees would be in the forest now.

I found the Introduction well written and clear. The description of the sites in the Material and Methods section can be improved. But the Results section makes the whole manuscript unsuited for publication.

The main problem is that there are no statistical analysis whatsoever. So all the hypotheses made remain unverified. Authors describe a series of figures which illustrate several pattern with time that I could not see and with no statistical validation. Authors keep writing that tree size heterogeneity and amount of dead wood increase with time from last intervention but from the pictures I could only see “U” shape patterns (Fig 3 and 4) or absence of clear pattern (Fig 5, 6 and 7). The absence of any statistical test makes their description open to argument and many results seems very easy to object. Most figures show bars with mean and standard deviations related to different periods of times, but bars seem to overlap a lot with each others and there is no possibility to check if different periods of time show significantly different mean values of structural diversity or amount of dead wood or number of large trees, etc.

Moreover the Results section is way too long (14 full pages) and repetitive. This section was very hard to read and need to be shortened considerably.

I would suggest authors to simplify their results and make at least some easy statistical regression tests using time since intervention as explanatory variable and the same measures used in the manuscript as response variable. I would also suggest not to group data in decades but to make a much more powerful test just using every plot with its time since intervention and it Gini index, or amount of dead wood etc. I would try also to include some quadratic term for time in the regression to take care of non linearity (U shape) or to use GAMs. Authors should also consider that these plot data are probably spatially autocorrelated and their analysis should take care of this problem too.

I found also some minor problems:

- there is no a clear definition of “set aside” in the introduction. I think it would considerably clarify the text

- the legend of Figure 1 is not clear. Some colors are very easy to mistake

- row 121 use “continuous” instead of “compact”

- I do not understand the difference between description of sites given in rows 131-139 from the one given in 117-123. Please clarify or delete.

- equation no. 2 is not correctly written (some log do not have brackets)

- why Figure 2 include a buffer area. Please clarify

- description of forest history from row 245 to row 265 should go in Material and Methods

- caption of figure 3 seems incorrect. Shouldn’t “a)and “b) be exchanged?

- the text starting with “*” and “**” at the and of Table 2 should go in material and methods and be described better

- several bibliographic references are incomplete (ex: in ref 23, 24, 30 and 45 the journal is missing)

Author Response

Dear Reviewer,

Thank you very much for your constructive comments. We addressed every comment with appropriate changes in the paper and we justified below these changes.

Reviewer comment

Authors response

1.

The manuscript aims to analyze the set-aside impact on the current diversity of mountain temperate forests from the Natura 2000 of the Rumanian site Rarău-Giumalău. The dataset is important since they analyze some sites with no past evidence of human intervention and several other sites where timber harvesting was done several decades ago. The main objective is to study the effect of time elapsed from last human impact (or time from strict reserve establishment) on structural diversity (measured by Gini index on basal area), on the amount of dead wood still in the forest, on large tree density and on regeneration. T and more dead wood, the more large trees would be in the forest now.

We agree that the general expectation is that the more time has passed since last human intervention, the more diverse the structural diversity of the site should be. However, the analyzed site is quite complex from both perspectives: ecological and of the management applied in the past. This is way the results do not seem spectacular, but the trends we identified reveal how the diversity depends in the study area on the past forest management.

2.

I found the Introduction well written and clear. The description of the sites in the Material and Methods section can be improved. But the Results section makes the whole manuscript unsuited for publication.

We improved the „Material and Methods” and the „Results” sections, considering your suggestions.

3.

The main problem is that there are no statistical analysis whatsoever. So all the hypotheses made remain unverified. Authors describe a series of figures which illustrate several pattern with time that I could not see and with no statistical validation. Authors keep writing that tree size heterogeneity and amount of dead wood increase with time from last intervention but from the pictures I could only see “U” shape patterns (Fig 3 and 4) or absence of clear pattern (Fig 5, 6 and 7). The absence of any statistical test makes their description open to argument and many results seems very easy to object. Most figures show bars with mean and standard deviations related to different periods of times, but bars seem to overlap a lot with each others and there is no possibility to check if different periods of time show significantly different mean values of structural diversity or amount of dead wood or number of large trees, etc.

We thank you very much for this suggestion! We improved the „Methods” and „Results” section with statistical analysis and adjusted accordingly both the graphs and explanatory text.

The "U" shape appears in the graphs due to the inclusion in the analysis of stands that were never under strictly protection regime, but which have always been managed through a close-to-nature forestry system. On the other hand, some of the stands included in the strictly protection regime were managed intensively. These aspects have been explained in the manuscript. If we have had excluded these stands, the trend lines would have been more evident, but the analysis would have been uncompleted.

4.

Moreover, the Results section is way too long (14 full pages) and repetitive. This section was very hard to read and need to be shortened considerably.

The paper was reorganized, considering the suggestions of both reviewers and the „Results” section has 6 pages. It had 7 pages in the previous form of the manuscript.

5.

I would suggest authors to simplify their results and make at least some easy statistical regression tests using time since intervention as explanatory variable and the same measures used in the manuscript as response variable. I would also suggest not to group data in decades but to make a much more powerful test just using every plot with its time since intervention and it Gini index, or amount of dead wood etc. I would try also to include some quadratic term for time in the regression to take care of non linearity (U shape) or to use GAMs. Authors should also consider that these plot data are probably spatially autocorrelated and their analysis should take care of this problem too.

We considered the suggestion and used statistical tests to verify the significance of the differences. We analyzed the data both in the "grouped into classes" and in the "ungrouped into classes" version, and since the results obtained were very similar, we kept in the manuscript the analysis version with values grouped into classes. The analysis of the differences among means was carried out using the Kruskal–Wallis nonparametric test (because all experimental distributions differ from the normal distribution (Shapiro–Wilk test), even after the data transformation (x’ = log (x + 1)). When significant differences were found, we used the Conover–Iman procedure for multiple pairwise comparison. We started our analysis with all 319 SPs installed in the forest habitats from the study area, but after that, in order to reduce the influence of spatial autocorrelation, we focused only on SPs from 1-PRP (500 x 500 m square grid). This aspect is mentioned in the manuscript.

6

.

I found also some minor problems:

- there is no a clear definition of “set aside” in the introduction. I think it would considerably clarify the text

A definition for „set aside forests” was included in the Introduction:

„Set-aside forests can be defined as lands covered by forests primarily managed for the purpose of nature conservation”.

- the legend of Figure 1 is not clear. Some colors are very easy to mistake

We modified Figure 1 according to suggestion and replaced it in the manuscript.

- row 121 use “continuous” instead of “compact”

We changed “compact” to “continuous”.

- I do not understand the difference between description of sites given in rows 131-139 from the one given in 117-123. Please clarify or delete.

Lines 131-139 complete the description of the study area given in lines 117-123, referring only to the nature and scientific reserves inside the Natura 2000 site. These reserves were established long before the establishment of Rarău-Giumalău Natura 2000 site. For clarification, we added in L117-119 the year of Rarău-Giumalău establishment as Natura 2000 site.

- equation no. 2 is not correctly written (some log do not have brackets)

Brackets were added.

- why Figure 2 include a buffer area. Please clarify

Explanations added in section 2.2:

-        For Slătioara reserve: „In 1950 a buffer area of 360 ha was established, being maintained till 2006, when it was included in strictly protection regime”;

-        For Giumalău reserve: „In this case, part of the reserve was transformed in buffer area from 1983 to 1994, due to the damages produced by wind in some forest compartments and to that time decision to extract the damaged wood. As in case of Slătioara reserve, low intensity cuts were allowed in buffer areas, aiming to isolate the core area from external influences”.

- description of forest history from row 245 to row 265 should go in Material and Methods

The entire Section 3.1 was moved to ”Material and methods”, as section 2.2.

- caption of figure 3 seems incorrect. Shouldn’t “a)” and “b)” be exchanged?

Figure 3 is correct. The values on second vertical axis show the differences between all SPs in figure 3-a and SPs from 1-PRP in figure 3-b.

- the text starting with “*” and “**” at the end of Table 2 should go in material and methods and be described better

We moved the indicated text in „Data collection” section and described better:

„Stand structure is established in FMPs according to a scale from 1 to 4 (1 – even-sized; 2 – two-sized; 3 – uneven-sized irregular; 4 – uneven-sized balanced), where the value “1” corresponds to homogenous stands, while “4” indicates heterogeneous stands. Stand density is quantified in FMPs based on a scale between 0 (trees are lacking) and 1 (the projection of tree crowns to the ground covers entirely the stand area), with a 0.1 step”.

- several bibliographic references are incomplete (ex: in ref 23, 24, 30 and 45 the journal is missing)

We revised the bibliographic references 22, 23, 25, 30, 45, 62, 63, 65, 66, 73, 74 and 75. Reference 24 was correct.

The numbering of bibliographic references corresponds to the previous version of the manuscript, which was analyzed by reviewers.
